_Article_

# Liver ACSM3 deficiency mediates metabolic syndrome via a lauric acid-HNF4α-p38 MAPK axis

Xiao Xiao[1,3], Ruofei Li[1,3], Bing Cui[1,3], Cheng Lv[1], Yu Zhang[1], Jun Zheng[2], Rutai Hui[1] & Yibo Wang [1✉]

## Abstract

**Metabolic syndrome combines major risk factors for cardiovascular disease, making deeper insight into its pathogenesis important. We here explore the mechanistic basis of metabolic syndrome by recruiting an essential patient cohort and performing extensive gene expression profiling. The mitochondrial fatty acid metabolism enzyme acyl-CoA synthetase medium-chain family member 3 (ACSM3) was identified to be significantly lower expressed in the peripheral blood of metabolic syndrome patients. In line, hepatic ACSM3 expression was decreased in mice with metabolic syndrome. Furthermore, _Acsm3_ knockout mice showed glucose and lipid metabolic abnormalities, and hepatic accumulation of the ACSM3 fatty acid substrate lauric acid. Acsm3 depletion markedly decreased mitochondrial function and stimulated signaling via the p38 MAPK pathway cascade. Consistently, _Acsm3_ knockout mouse exhibited abnormal mitochondrial morphology, decreased ATP contents, and enhanced ROS levels in their livers. Mechanistically, _Acsm3_ deficiency, and lauric acid accumulation activated nuclear receptor Hnf4α-p38 MAPK signaling. In line, the p38 inhibitor Adezmapimod effectively rescued the Acsm3 depletion phenotype. Together, these findings show that disease-associated loss of ACSM3 facilitates mitochondrial dysfunction via a lauric acid-HNF4a-p38 MAPK axis, suggesting a novel therapeutic vulnerability in systemic metabolic dysfunction.**

**Keywords** Metabolic Syndrome; Acsm3; Mitochondria Dysfunction; Lauric Acid; p38 MAPK Signaling Pathway
**Subject Categories** Metabolism; Molecular Biology of Disease

## Introduction

Metabolic syndrome (MetS) accounts for the largest burden of non-communicable diseases worldwide (Bishehsari et al, 2020). In epidemiological studies, MetS occurrence varies between 20 and 45% of the population and is expected to increase to ~53% by 2035 (Engin, 2017; Gierach et al, 2014). MetS refers to the co-occurrence of several known cardiovascular risk factors, including hypergly-cemia/insulin resistance (IR), obesity, atherogenic dyslipidemia,

and hypertension (Huang, 2009; Mensah et al, 2019; Roth et al, 2020). These conditions are interrelated and share underlying mediators, mechanisms, and pathways (Cornier et al, 2008; Després and Lemieux, 2006).

In MetS, the increased flux of fatty acids (FA) in hepatocytes leads to enhanced mitochondrial FA import and oxidation. FA metabolism mainly occurs in mitochondria, which are abundant in the liver (Bhatti et al, 2017). Mitochondria are the intracellular double membrane-bound organelles and play crucial roles in metabolizing nutrients, producing adenosine triphosphate (ATP), and responding to numerous processes such as energy metabolism, free radicals generation, and cellular homeostasis, including in the regulation of cellular respiration, oxidative phosphorylation, and reactive oxygen species (ROS) balance (Bhatti et al, 2017; Kastaniotis et al, 2017; Wallace et al, 2010). Remarkably, mitochondrial β-oxidation is the major pathway for the degradation of FA, a breakdown of this process results in the excess accumulation of FAs in eukaryotes, incorporated into triglycerides (or triacylglycerols), phospholipids as well as into other lipid species, which contribute to the progress of metabolic disorders (Houten et al, 2016; Ren et al, 2010; Yoon et al, 2021). In addition, considerable research indicates that decreased FA oxidation impairs insulin signaling, which consequently results in free FA and IR, further reducing mitochondrial oxidative capacity and ATP synthesis (Amorim et al, 2010; Koves et al, 2008; Sim et al, 2019; Turner and Heilbronn, 2008).

In this study, we first recruited an essential MetS cohort with strict criteria and performed genome-wide transcriptome analysis. _ACSM3_ (acyl-CoA medium-chain family member 3) was identified to be significantly lower expressed in the peripheral blood of MetS patients, which was validated in another larger cohort. Furthermore, Acsm3 was also markedly decreased in the peripheral blood of MetS mice, compared with controls. The tissue distribution of Acsm3 was analyzed in mice, and high expression was found in the liver, which is an important organ in glucose and lipid metabolism. Simultaneously, hepatic Acsm3 was also downregulated in MetS mice, consistent with those in the peripheral blood. ACSM3 is a member of the acyl-CoA medium-chain synthetase (ACSM) family and is localized on the outer membrane of mitochondria, which catalyzes the activation of medium-chain (C4-C14) length FAs and xenobiotic carboxylic acids (Watkins et al, 2007). In previous studies, ACSM3 has been confirmed to be associated with several types of carcinomas (Yang et al, 2022; Zhao et al, 2022). Nevertheless, its participation in MetS is still unclear. Herein, we

[1]State Key Laboratory of Cardiovascular Disease, Fuwai Hospital, National Center for Cardiovascular Diseases, Chinese Academy of Medical Sciences and Peking Union Medical College, Beijing, China. [2]Rizhao Port Hospital, Shandong, China. [3]These authors contributed equally: Xiao Xiao, Ruofei Li, Bing Cui. ✉E-mail: yibowang@hotmail.com

determined the lower expression of ACSM3 in MetS. Thus, we hypothesized that ACSM3 is involved in metabolic disorders and may serve as a therapeutic target of MetS. We explored the underlying mechanisms in this study.

# Results

## Lower *ACSM3* expression was observed in MetS patients and mice

A small cohort with strict inclusion criteria of MetS was used for expression profile chip analysis. A total of 69 subjects, including 51 MetS patients and 18 controls, were selected from an existing cohort in Rizhao Port Hospital in Shandong, China. The original cohort in Rizhao Port Hospital was established in 2010 for a hypertension prevention study. All participants were from Rizhao Port Company and lived in Rizhao, Shandong, China. The cohort was followed until 2013. MetS patients and controls were eligible in 2010 and in the whole follow-up period (Cui et al, 2021; Li et al, 2014). The expression profile chip results showed that *ACSM3* was lower in the peripheral blood of MetS patients, with the largest fold difference (Fig. 1A). RT-qPCR assays further verified the lower expression of *ACSM3* in MetS patients (Fig. 1B).

The second validation cohort was selected from a cohort established for a resistant hypertension study in China. Subjects aged 60–75 years were selected for the validation study, including 826 individuals from 2012 to 2015 (Wu et al, 2018). Among them, 386 subjects were classified with MetS, and 440 were control subjects. 48.2% (212/440) of control individuals were male. 51.8% (200/386) of MetS patients were male. The expression of *ACSM3* was lower in the MetS group in the second validation cohort (Fig. 1C).

Consistent with the above two cohorts, we also found that Acsm3 was markedly downregulated in the peripheral blood of MetS mice, compared with those in the controls (Fig. 1D). We then analyzed the tissue distribution of Acsm3 in mice, and high expression was found in the liver (Figs. 1E,F and EV1A,B). The liver is one of the most crucial places for glucose and lipid metabolism, and its molecular anomalies are closely related to MetS (Trefts et al, 2017). We further assayed hepatic Acsm3 expression, and the results showed that it was also markedly decreased in MetS mice (Fig. 1G,H).

In summary, our data demonstrated that ACSM3 was lower expressed in MetS patients and mice.

## Loss of Acsm3 resulted in MetS and promoted hepatic FA accumulation, especially lauric acid not only in systemic knockout mice but also in liver-specific knockout mice

ACSM3 is an acyl-CoA synthetase that takes part in the first step of FA metabolism. To further address its role in MetS, the phenotype of Acsm3 knockout mice was analyzed with wild-type littermates as controls. *Acsm3* systemic knockout mice were generated by the CRISPR/Cas9 system (Fig. 2A) and exhibited profound depletion of Acsm3 expression in liver extracts (Fig. 2B). When continuously fed with ND for 12 weeks (Fig. 2C), the body weight was lower, while the liver weight and liver/body weight ratio were higher when compared with controls (Fig. 2D). The reduced body weight in knockout mice might be due to a reduction in absolute food intake.

Averaged daily feed intakes of the knockout mice were lower (Fig. 2E). To measure the effects of *Acsm3* on glucose homeostasis, insulin tolerance tests (ITT), glucose tolerance tests (GTT), glucose and insulin measurements, and homeostasis model assessment of insulin resistance (HOMA-IR, = fasting Glu (mmol/L) × fasting Ins (μU/mL)/22.5) calculations were conducted. *Acsm3* knockout mice showed impaired glucose homeostasis with larger glucose excursion in GTT (Fig. 2F), decreased insulin sensitivity in ITT (Fig. 2G), and higher HOMA-IR (Fig. 2H–J) compared to control mice. However, there were no significant differences in serum ALT and AST contents between control and knockout mice under ND, suggesting that Acsm3 deletion alone cannot induce obvious hepatic damage (Fig. 2K). Next, we observed the effect of *Acsm3* knockout on hepatic lipid metabolism under ND. Hepatic and serum TG levels were significantly higher in the MetS group (Fig. 2L,M). However, there were no significant alterations in hepatic TC, NEFA, or serum TC, NEFA, HDL-C, and LDL-C levels when fed with ND (Fig. 2L,M). The metabolic phenotypes of the mice differed little under ND, so the mice were fed an FF diet from 4 to 12 weeks (Fig. EV2A), with the expectation that this would amplify differences between the *Acsm3* knockout and control groups.

When fed FF, *Acsm3* knockout mice also presented lower body weight, higher liver weight, and a higher liver/body weight ratio (Fig. 3A). Averaged daily feed intakes of the knockout mice were lower (Fig. EV2B). The *Acsm3* knockout mice displayed an impaired insulin response as determined by GTT, ITT, and HOMA-IR (Figs. 3B–D and EV2C,D). Serum ALT and AST levels were both increased after Acsm3 deletion under the FF diet (Fig. 3E). Significant increases in hepatocellular ballooning degeneration and lipid accumulation in *Acsm3* knockout mice were observed by H&E and oil red O staining, respectively (Fig. 3F,G). Serum ALT, AST, TC, TG, NEFA, HDL-C, LDL-C, or hepatic TC, TG, and NEFA levels were all elevated in *Acsm3* knockout mice (Figs. 3H and EV2E). Furthermore, we found that the acetyl-CoA level was significantly decreased in the knockout mice fed an FF diet (Fig. 3I).

The MetS phenotype was detected in whole-body *Acsm3* knockout mice, but to be more rigorous, we created a liver-specific virus to exclusively knock down Acsm3 in the liver and found similar results (Figs. EV3A–J and EV4A–J). Liver-specific Acsm3 knockdown mice revealed mild metabolic abnormalities with ND, consistent with the phenotypes caused by the systematic Acsm3 knockout mice (Fig. 2A–M). After hepatic Acsm3 deficiency, the mice exhibited impaired glucose tolerance and insulin tolerance and increased HOMA-IR indexes (Fig. EV3D–G). In addition, their hepatic and serum TG contents were also upregulated (Fig. EV3I,J). After 12 weeks of FF feeding, the hepatic knockdown mice showed a more severe MetS phenotype similar to systematic Acsm3 deletion, which illustrated the crucial role of hepatic Acsm3 in promoting MetS (Figs. 3A–H and EV4C–J).

Since *Acsm3* deletion in hepatocytes aggravated lipid accumulation in the FF-induced model, we also performed FA composition analysis by mass spectrometry ($n = 6$ mice in each group). In both the ND and FF groups, all detectable medium-chain FAs were upregulated to varying degrees in *Acsm3* knockout mice, with the most prominent upregulation occurring in C12, i.e., lauric acid (Fig. 3J,K).

Taken together, loss of Acsm3 resulted in MetS and significantly aggravated hepatic FA accumulation, especially lauric acid.

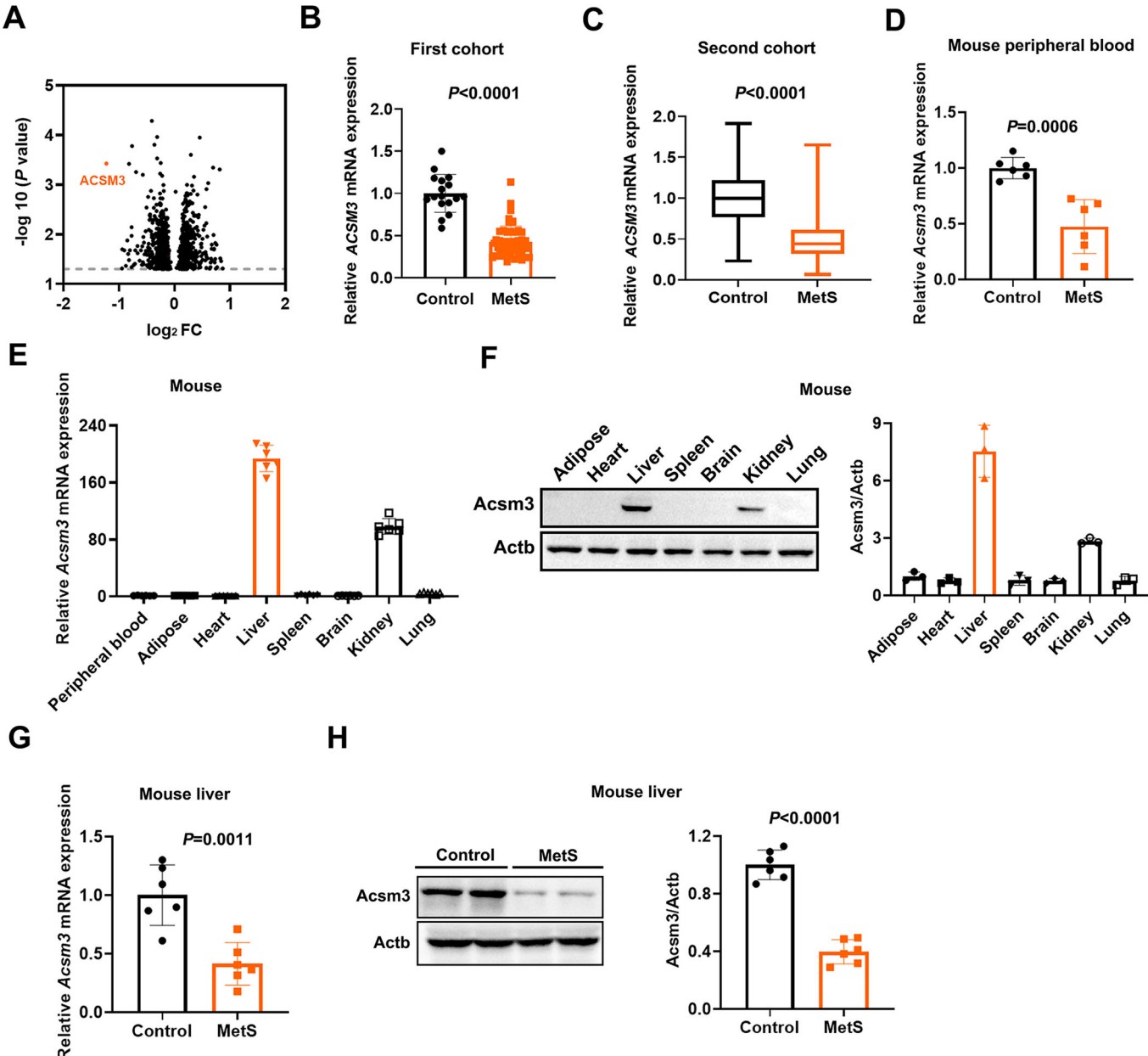

**Figure 1. ACSM3 expression in metabolic syndrome patients and mice.**

(**A**) Volcano plot showing the gene expression from whole blood cells between 18 healthy subjects and 51 metabolic syndrome (MetS) patients. *x* axis: log$_2$FC, Fold change (unpaired *t* test) displayed on a log2 scale. *y* axis: −log10 (*P* value). (**B**) The relative mRNA expression of *ACSM3* was verified in whole blood cells in the above cohort using a real-time quantitative PCR (RT–qPCR) assay. Values were represented as the mean ± SD. Statistics were performed using Student's *t* test. (**C**) The relative mRNA expression of *ACSM3* was verified in the second cohort (*n* = 440 subjects in the control group; *n* = 386 subjects in the MetS group). In the box plot, boxes indicate ranges from the first to third quartiles, and the bold central lines display the median. Upper and lower whiskers extend to the maximum or minimum values no further than 1.5 times the interquartile range. Statistics were performed using Student's *t* test. (**D**) The relative mRNA expression of *Acsm3* in the peripheral blood of control and MetS mice (*n* = 6 biologically independent samples in each group). Values were represented as the mean ± SD. Statistics were performed using Student's *t* test. (**E**) The relative mRNA expression of *Acsm3* in different tissues of mice (*n* = 6 biologically independent samples in each group). Values were represented as the mean ± SD. (**F**) Western blot and quantification of the relative protein expression of Acsm3 in different tissues of mice. Three results were used for quantification, *n* = 3 biologically independent samples in each group. Values were represented as the mean ± SD. (**G**) The relative mRNA expression of *Acsm3* in the livers of control and MetS mice (*n* = 6 biologically independent samples in each group). Values were represented as the mean ± SD. Statistics were performed using Student's *t* test. (**H**) Western blot and quantification of the relative protein expression of Acsm3 in the livers of control and MetS mice. Three results were used for quantification, *n* = 6 biologically independent samples in each group. Values were represented as the mean ± SD. Statistics were performed using Student's *t* test. Source data are available online for this figure.

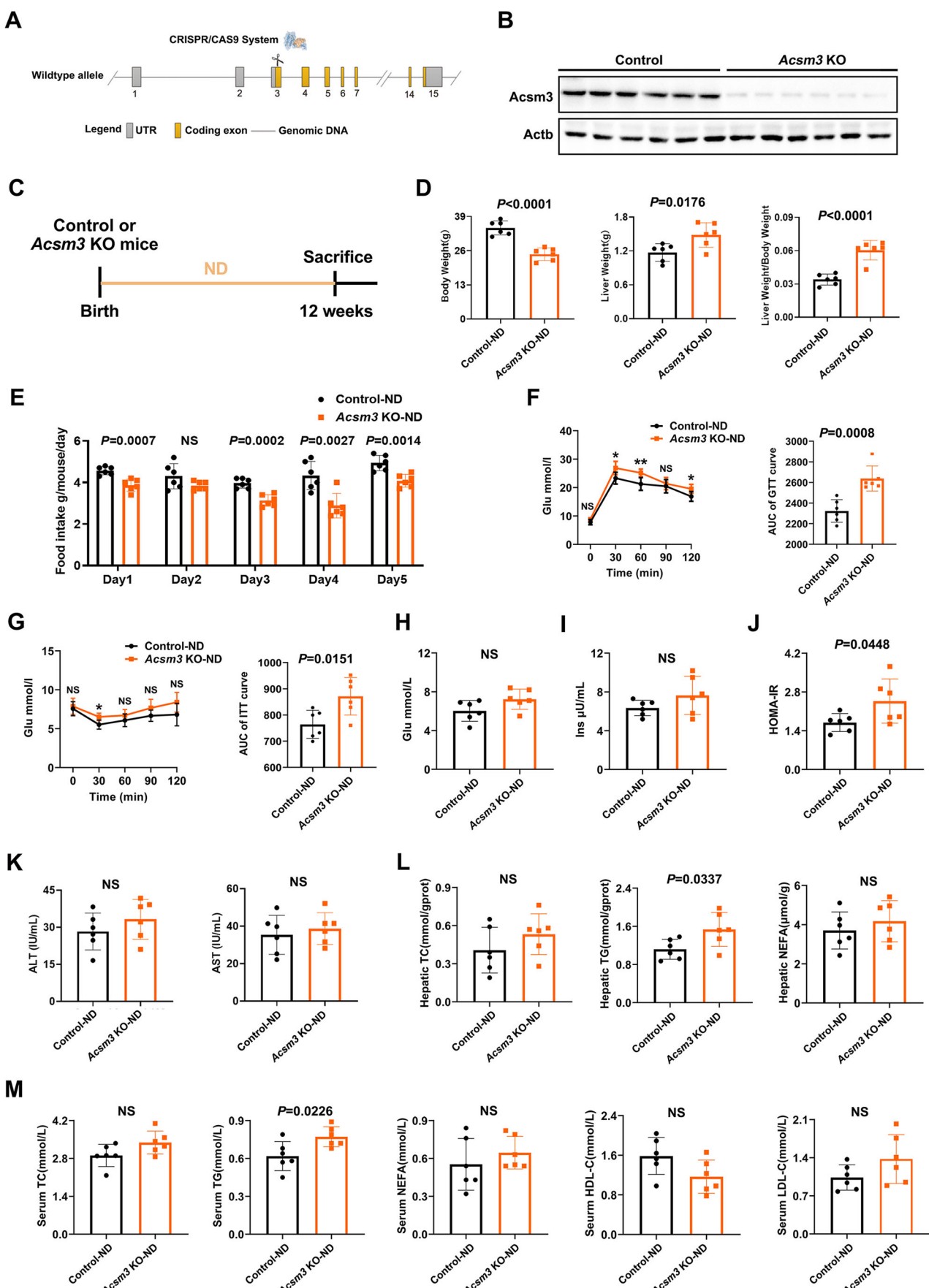

◄ **Figure 2. *Acsm3* knockout mice under normal diet.**

(A) Construction of *Acsm3* knockout (KO) mice by the CRISPR/Cas9 system. (B) Hepatic Acsm3 protein expression was detected by western blot ($n = 6$ biologically independent samples in each group). (C) Timeline of *Acsm3* KO and control mice fed with ND. (D) The body weights (g), liver weights (g), and liver weight/body weight of *Acsm3* KO and control mice ($n = 6$ biologically independent samples in each group). Values were represented as the mean ± SD. Statistics were performed using Student's *t* test. (E) The average daily food intake of *Acsm3* KO and control mice for five consecutive days ($n = 6$ biologically independent samples in each group). Values were represented as the mean ± SD. Statistics were performed using Student's *t* test. (F, G) Glucose tolerance test (GTT), insulin tolerance test (ITT), and their respective area under the curve (AUC) between *Acsm3* KO and control mice ($n = 6$ biologically independent samples in each group). Values were represented as the mean ± SD. Statistics were performed using Student's *t* test. *P* values were denoted by asterisks: *$P < 0.05$, **$P < 0.01$. (H–J) The fasting Glu (mmol/L), fasting Ins (μU/mL), and homeostasis model assessment of insulin resistance (HOMA-IR) indexes of *Acsm3* KO and control mice ($n = 6$ biologically independent samples in each group). HOMA-IR = fasting Glu (mmol/L) × fasting Ins (μU/mL)/22.5. Values were represented as the mean ± SD. Statistics were performed using Student's *t* test. (K) The contents (IU/mL) of serum ALT and AST in *Acsm3* KO and control mice ($n = 6$ biologically independent samples in each group). Values were represented as the mean ± SD. Statistics were performed using Student's *t* test. (L) The contents of hepatic TC (mmol/gprot), TG (mmol/gprot), and NEFA (μmol/g) in *Acsm3* KO and control mice ($n = 6$ biologically independent samples in each group). Values were represented as the mean ± SD. Statistics were performed using Student's *t* test. (M) The contents (mmol/L) of serum TC, TG, NEFA, HDL-C, and LDL-C in *Acsm3* KO and control mice ($n = 6$ biologically independent samples in each group). Values were represented as the mean ± SD. Statistics were performed using Student's *t* test. Source data are available online for this figure.

## Loss of Acsm3 impaired hepatic mitochondrial morphology and aggravated mitochondrial dysfunction

To investigate the underlying mechanisms further, liver tissues in control and knockout mice fed with FF were subjected to transcriptome analysis. The results for GO pathway enrichment of DEGs showed that many metabolic-related signaling pathways were involved (Fig. 4A, $n = 3$ in each group), such as "fatty acid metabolic process" and "lipid oxidation". Notably, the DEGs were significantly enriched in signaling pathways associated with mitochondrial function, such as "mitochondrial membrane organization", "regulation of mitochondrial membrane permeability", and "mitochondrial ATP synthesis coupled proton transport" (Fig. 4A), suggesting that *Acsm3* deletion may be associated with mitochondrial function.

Based on this, we conducted further experimental validation. To determine whether *Acsm3* knockout affects mitochondrial morphology, we used transmission electron microscopy (TEM) to examine mitochondrial ultrastructure. We quantified the cross-sectional area and perimeter of mitochondria using Image-Pro Plus 6.0. Electron microscopy revealed more abnormal mitochondrial morphology in *Acsm3* knockout mice fed with FF (Fig. 4B), along with significantly increased mitochondrial perimeter and area (Fig. 4C,D).

Hepatic mitochondrial dysfunction leads to the derangement of mitochondrial electron transport chain activities, energy production, and impaired lipid metabolism (Diao et al, 2018). We first determined the metabolic activity of these mice by measuring their respiratory quotient (respiratory exchange ratio) using Oxymax metabolic activity cages. The results showed that the energy expenditure of the knockout mice showed a lower respiratory quotient ($RQ = VCO_2/VO_2$) than that of the control mice (Fig. 4E). To quantify changes in oxidative metabolism in mitochondria, we further measured the oxygen consumption rate (OCR) of primary mouse hepatocytes. Primary hepatocytes of the knockout mice displayed decreased OCR compared with control mouse hepatocytes, indicating damage to mitochondrial oxidative metabolism in the *Acsm3* knockout hepatocytes (Fig. 4F).

Mitochondria are the "powerhouse" of cells, as they are the main site of energy currency ATP production. The mitochondrial electron transport chain (ETC) is the main site of ATP generation, ETC activity may result in excessive electron leakage and thus excessive ROS generation and cellular injury (Prasun et al, 2021).

To determine whether Acsm3 deletion leads to mitochondrial dysfunction in hepatocytes, we examined total and mitochondrial ROS levels. The results showed that *Acsm3* knockout hepatocytes displayed higher levels of total and mitochondrial ROS than control hepatocytes (Fig. 4G,H). The accumulation of ROS will lead to decreased mitochondrial membrane potential and ATP production (Zorov et al, 2014). The knockout hepatocytes exhibited decreased intracellular ATP levels and mitochondrial membrane potential (Fig. 4I,J). Thus, *Acsm3* deletion impaired mitochondrial morphology and aggravated hepatic mitochondrial dysfunction.

## Mitochondrial dysfunction was mediated by the C12-HNF4A-P38 MAPK signaling pathway

The results for GO pathway enrichment of DEGs of transcriptome analysis were also significantly enriched in the p38 MAPK cascade (Fig. 5A). P38 mitogen-activated protein kinase (p38 MAPK) signaling is closely related to a variety of intracellular responses, including inflammation, oxidative stress, ROS, and apoptosis, which can participate in MetS by multiple approaches (Chan et al, 2019; Woo et al, 2023). Noticeably, previous studies have shown that p38 MAPK (MAPK14) is a crucial player in mitochondrial dysfunction (Huang et al, 2018; Huang et al, 2013; Manikanta et al, 2020). Excessive activation of the p38 MAPK pathway results in disrupting mitochondrial membrane potential and releasing ROS(Huang et al, 2018). Inhibition of the p38 MAPK pathway could attenuate mitochondrial dysfunction (Huang et al, 2018; Kamoshita et al, 2022; Kumphune et al, 2015). Therefore, we hypothesized that p38 MAPK may be a vital factor involved in hepatic mitochondrial dysfunction and MetS in Acsm3 knockout mice. By analyzing the individual gene expression included in the p38 MAPK cascade of the GO pathway enrichment (Fig. 5A), it can be seen that the p38 MAPK pathway was significantly activated in *Acsm3* knockout mice. The transcriptome results of *Mapk14* were confirmed by RT-qPCR assay (Fig. 5B). As shown in Fig. 5C, the expression levels of p38 MAPK and phosphorylated/total p38 MAPK protein levels were significantly increased in *Acsm3* knockout mice. These results supported the idea that *Acsm3* knockout mouse-induced mitochondrial dysfunction could occur through the p38 MAPK signaling pathway.

However, the mechanism that how *Acsm3* knockout caused the upregulation of p38 MAPK was not clear. Since FAs composition analysis by mass spectrometry indicated that lauric acid was the

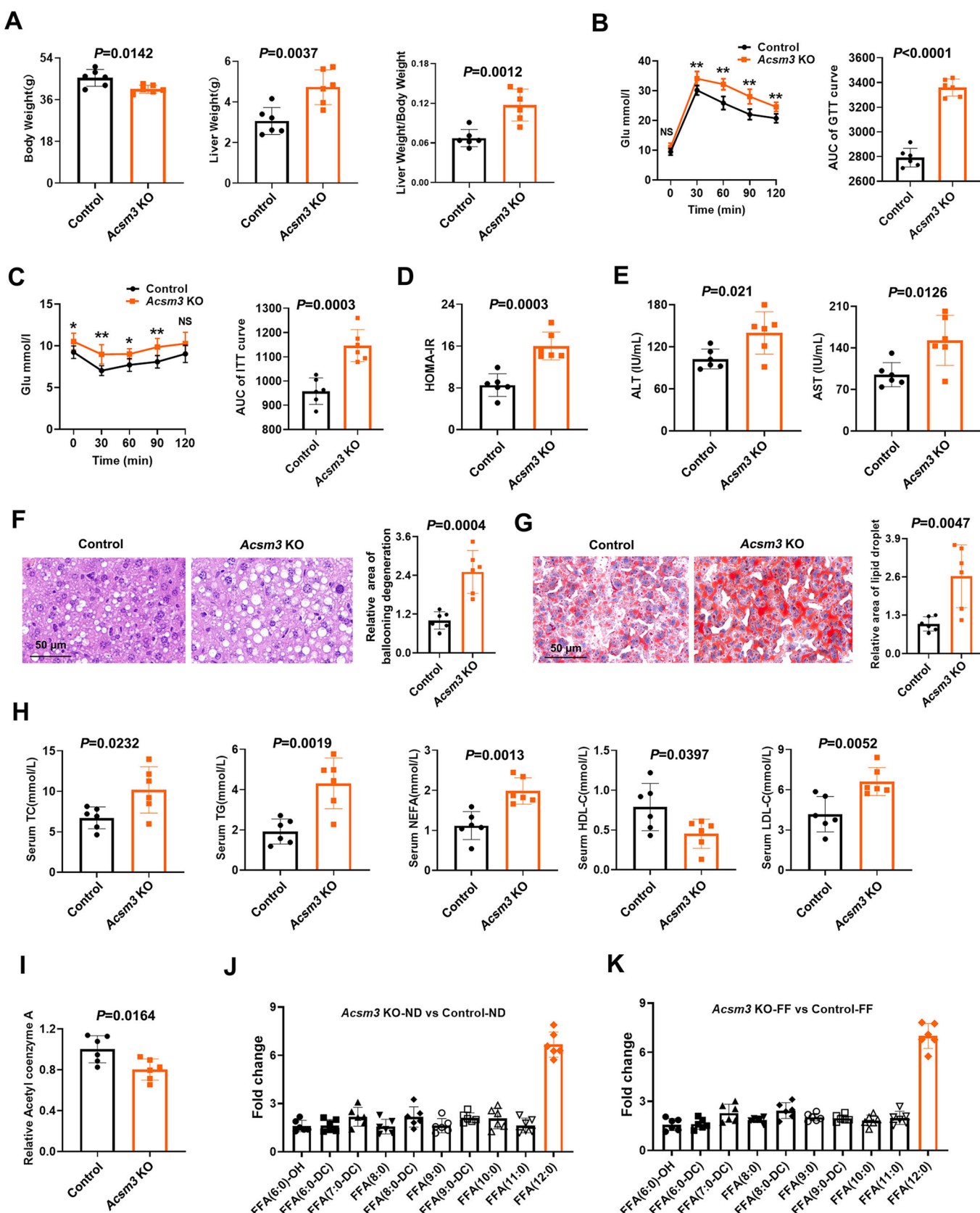

**Figure 3. *Acsm3* knockout mice involved in metabolic syndrome.**

All mice in this figure (except (J)) were fed with an FF diet for 8 weeks. (A) The body weights (g), liver weights (g), and liver weight/body weight of *Acsm3* KO and control mice ($n = 6$ biologically independent samples in each group). Values were represented as the mean ± SD. Statistics were performed using Student's *t* test. (B, C) GTT, ITT, and their respective AUC of *Acsm3* KO and control mice ($n = 6$ biologically independent samples in each group). Values were represented as the mean ± SD. Statistics were performed using Student's *t* test. *P* values were denoted by asterisks: *$P < 0.05$, **$P < 0.01$. (D) HOMA-IR indexes of *Acsm3* KO and control mice ($n = 6$ biologically independent samples in each group). HOMA-IR = fasting blood glucose (Glu) (mmol/L) × fasting insulin (Ins) (μU/mL)/22.5. Values were represented as the mean ± SD. Statistics were performed using Student's *t* test. (E) The contents (IU/mL) of serum ALT and AST in *Acsm3* KO and control mice ($n = 6$ biologically independent samples in each group). Values were represented as the mean ± SD. Statistics were performed using Student's *t* test. (F, G) Representative images of H&E and oil red O staining of *Acsm3* KO and control mouse livers after feeding FF diet for 8 weeks. Scale bar, 50 μm. The quantification of the relative area of ballooning degeneration or lipid droplets was based on NIH ImageJ software ($n = 6$ biologically independent samples in each group). Values were represented as the mean ± SD. Statistics were performed using Student's *t* test. (H) The contents (mmol/L) of serum total cholesterol (TC), triglyceride (TG), nonesterified fatty acid (NEFA), high-density lipoprotein cholesterol (HDL-C), and low-density lipoprotein cholesterol (LDL-C) in *Acsm3* KO and control mice ($n = 6$ biologically independent samples in each group). Values were represented as the mean ± SD. Statistics were performed using Student's *t* test. (I) The relative contents of acetyl-coenzyme A (Acetyl-CoA) in the livers of *Acsm3* KO and control mice ($n = 6$ biologically independent samples in each group). Values were represented as the mean ± SD. Statistics were performed using Student's *t* test. (J, K) Medium-chain fatty acids in the livers of *Acsm3* KO and control mice were detected by mass spectrometry. The data show the fold changes (*Acsm3* KO vs. control mice, under normal diet (ND) and FF diet, respectively). Source data are available online for this figure.

most prominent upregulated in *Acsm3* knockout mice (Fig. 3J,K), we next explored whether the upregulation of p38 MAPK after *Acsm3* knockout was mediated by lauric acid. The results showed that p38 MAPK and phosphorylated/total p38 MAPK were evaluated in primary hepatocytes when stimulated with lauric acid (Fig. 5D), indicating that the accumulation of lauric acid after *Acsm3* knockout could induce the upregulation of p38 MAPK.

To further explore potential factors that cause the upregulation of p38 MAPK induced by lauric acid stimulation, we used the JASPER tool (http://jaspar.genereg.net/) to predict transcription factors that might bind to the promoter region of *Mapk14*. By JASPER tool prediction, we found that the presence of the transcription factor Hnf4α, previously reported to be upregulated by lauric acid stimulation(Kamoshita et al, 2022), had a high binding score in the *Mapk14* promoter region (Fig. 5E). Then, the ChIP assay confirmed the binding of Hnf4α to the *Mapk14* promoter (Fig. 5E).

Subsequently, transcriptome analysis and RT-qPCR assays demonstrated that Acsm3 deletion could significantly upregulate *Hnf4a* mRNA expression, as could lauric acid stimulation (Fig. 5A,F,G). At the protein level, western blot results showed that Hnf4α was also upregulated in *Acsm3* knockout mice or stimulated by lauric acid (Fig. 5H,I), suggesting that the accumulation of lauric acid after *Acsm3* knockout led to the upregulation of Hnf4α, but the reason is unclear. Surprisingly, we discovered that Hnf4α exhibited strong binding to its own promoter area using the JASPER tool prediction. This finding was confirmed by the ChIP assay (Fig. 5J). FAs are endogenous ligands for HNF4α, and HNF4α activity is enhanced by ligand binding(Dhe-Paganon et al, 2002), so a luciferase reporter gene experiment was designed to prove whether lauric acid can promote the binding activity of Hnf4α to its own promoter region. The results showed that lauric acid stimulation significantly enhanced reporter gene activation (Fig. 5K). Taken together, our data clearly demonstrate that lauric acid can act as a transcriptional activator of *Hnf4a*, thereby promoting the expression of both Hnf4α and Mapk14.

The next step was to determine whether the upregulation of Mapk14 by lauric acid was through Hnf4α. The results showed that the knockdown of Hnf4α by siRNA significantly decreased the expression of p38 MAPK and phosphorylated/total p38 MAPK. When Hnf4α was knocked down in primary hepatocytes treated

with lauric acid, the upregulation of p38 MAPK and phosphorylated/total p38 MAPK induced by lauric acid disappeared (Fig. 5L), as well as induced by *Ascm3* knockout (Fig. 5M). Thus, the upregulation of p38 MAPK caused by lauric acid was via Hnf4α.

## The p38 MAPK inhibitor adezmapimod could rescue mitochondrial dysregulation and improve the MetS phenotype in *Acsm3* knockout mice

These findings suggested that enhanced p38 MAPK activity might be required for mitochondrial dysfunction in *Acsm3* knockout mice. Adezmapimod, a highly specific p38 MAPK inhibitor, inhibits the catalytic activity of p38 MAPK by competitive binding in the ATP pocket (Gum et al, 1998; Leelahavanichkul et al, 2013). We, therefore, tested the ability of adezmapimod to prevent mitochondrial dysfunction in *Acsm3* knockout and control mice. To do this, mice received an intraperitoneal injection of adezmapimod (10 mg/kg) for 2 weeks (Leelahavanichkul et al, 2013; Shen et al, 2019) or an injection of the same amount of 1% DMSO (Fig. 6A). Twelve weeks later, we first detected the effect of adezmapimod on p38 MAPK activity, and the results showed that the inhibitor substantially reduced the phosphorylation level of p38 MAPK in the liver tissue of *Acsm3* knockout mice (Fig. 6B). There was no significant change in body weight, liver weight, or liver/body weight ratio after treatment with or without adezmapimod (Fig. EV5). Adezmapimod-treated Acsm3 knockout mice showed significantly higher glucose tolerance and improved insulin sensitivity (Fig. 6C–E). The results of H&E staining and oil red O staining showed that adezmapimod effectively reduced ballooning degeneration and lipid deposition in the liver tissue of *Acsm3* knockout mice (Fig. 6F,G). Serum TC, TG, NEFA, HDL-C, and LDL-C levels were all decreased in adezmapimod-treated *Acsm3* knockout mice (Fig. 6H).

Next, we determined the metabolic activity of these mice by measuring their RQ using Oxymax metabolic activity cages. The results showed that the energy expenditure of the adezmapimod group showed higher respiratory quotients in *Acsm3* knockout mice (Fig. 6I). To quantify changes in oxidative metabolism in mitochondria, we further measured the oxygen OCR of primary mouse hepatocytes. Primary hepatocytes of the adezmapimod group displayed increased OCR in *Acsm3* knockout mice (Fig. 6J), suggesting the improvement of mitochondrial oxidative

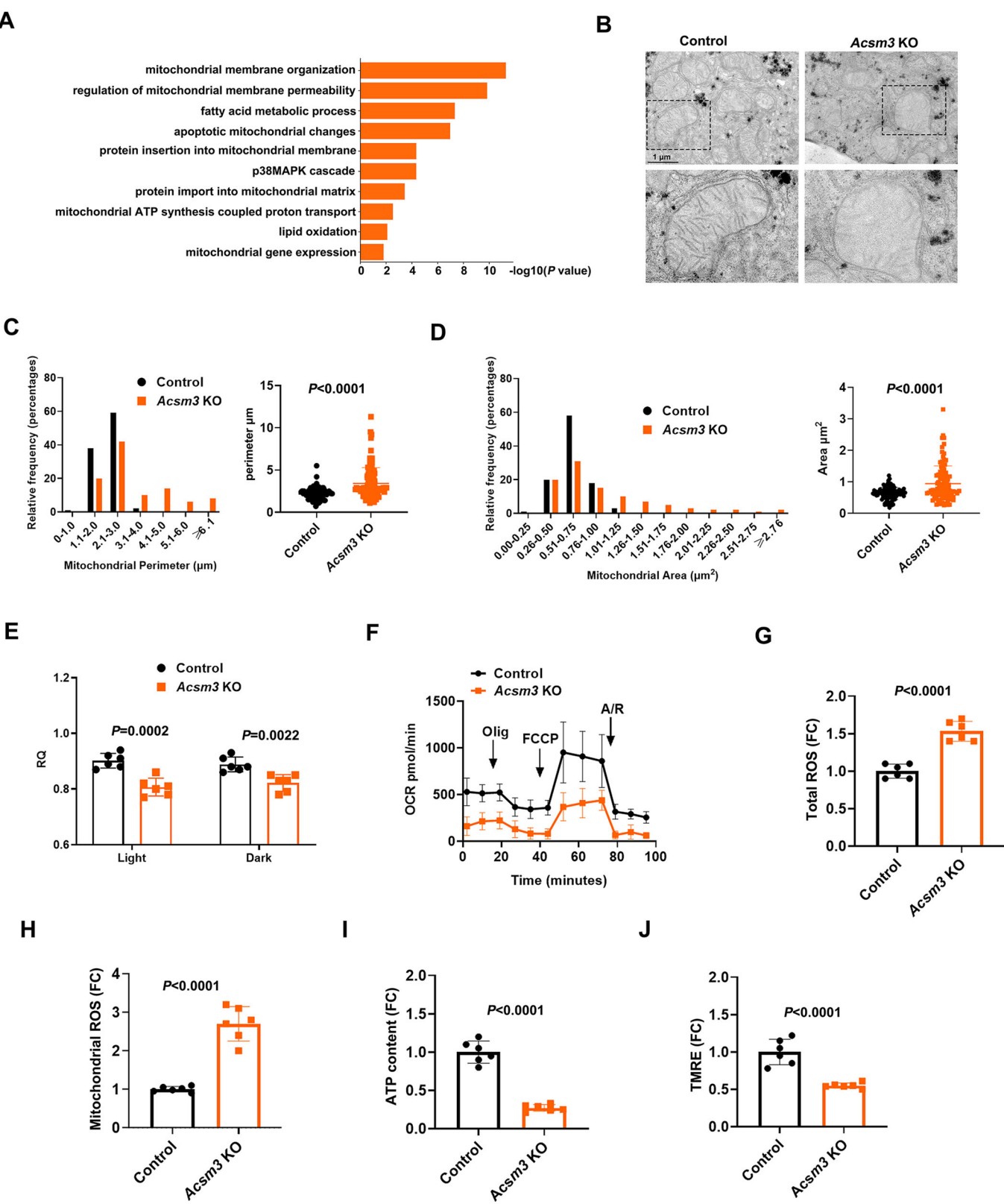

◄ **Figure 4.  The effect of _Acsm3_ knockout on mitochondrial function.**

All mice in this figure were fed with an FF diet for 8 weeks. (**A**) Bar plot showing the Gene Ontology (GO) enrichment analysis of the differentially expressed genes (DEGs) (unpaired _t_ test) in livers between _Acsm3_ KO and control mice. (**B**) Transmission electron microscopy (TEM) photographs of liver tissues from _Acsm3_ KO and control mice. Scale bar, 1 μm. (**C, D**) Distribution of the mitochondrial perimeter (μm) and mitochondrial area (μm$^2$) in the livers of _Acsm3_ KO and control mice ($n = 100$ mitochondria in each group for quantification). Values were represented as the mean ± SD. Statistics were performed using Student's _t_ test. (**E**) Respiratory quotient (RQ) of _Acsm3_ KO and control mice in light and dark, respectively ($n = 6$ biologically independent samples in each group). Values were represented as the mean ± SD. Statistics were performed using Student's _t_ test. (**F**) Oxygen consumption rate (OCR) analyses of primary hepatocytes perfused from _Acsm3_ KO and control mice. Oligomycin (Olig), Trifluoromethoxy carbonyl cyanide phenylhydrazone (FCCP), or antimycin/rotenone (A/R) treatments. ($n = 6$ biologically independent samples in each group). Values were represented as the mean ± SD. (**G, H**) Relative levels of total reactive oxygen species (ROS) and mitochondrial ROS in _Acsm3_ KO and control mouse livers ($n = 6$ biologically independent samples in each group). The data show the mean fluorescence intensities (MFIs). Values were represented as the mean ± SD. Statistics were performed using Student's _t_ test. (**I**) The relative contents of ATP in primary hepatocytes perfused from _Acsm3_ KO and control mice ($n = 6$ biologically independent samples in each group). Values were represented as the mean ± SD. Statistics were performed using Student's _t_ test. (**J**) Relative mitochondrial membrane potentials of the primary hepatocytes were detected using a TMRE probe ($n = 6$ biologically independent samples in each group). Data show the MFI. Values were represented as the mean ± SD. Statistics were performed using Student's _t_ test. Source data are available online for this figure.

metabolism in the adezmapimod group. In addition, total and mitochondrial ROS displayed lower levels in the adezmapimod-treated _Acsm3_ knockout group (Fig. 6K,L). Intracellular ATP levels and mitochondrial membrane potential were also increased in the adezmapimod group (Fig. 6M,N). These results supported that adezmapimod could rescue the mitochondrial dysregulation phenotype after _Acsm3_ knockout.

## Discussion

In this study, we identified ACSM3, a vital member of the ACSM family, whose deficiency promotes MetS progression. We determined that lauric acid was the specific FA catalyzed by Acsm3, and the mitochondrial dysfunction and metabolic disorders caused by Acsm3 were attributed to its induced activation of the lauric acid/Hnf4α/p38 MAPK pathway. Our study provided novel targets for mechanism decryption and clinical therapy of MetS (Fig. 6O).

Acyl-CoA synthetases (ACSs) are essential for de novo lipid synthesis, FA catabolism, and membrane remodeling (Soupene and Kuypers, 2008). ACSs perform the initial reaction for cellular FA metabolism by ligating a coenzyme A to an FA, which both traps an FA within a cell and activates it for metabolism (Fernandez and Ellis, 2020). The ACS family of enzymes is large and can be broken into subfamilies termed short-, medium-, long-, and very-long-chain ACSs, each with unique distribution across and within cell types and differential FA substrate preferences (Fernandez and Ellis, 2020; Watkins et al, 2007). ACSM3 is a member of the ACSM family, which was identified to be lower expressed in MetS patients and mice in our study. In a recent study, Junková et al compared the hepatic transcriptome of spontaneously hypertensive rats, polydactylous rats (PD, an animal model of hypertriglyceridemia, IR, and obesity), and Brown Norway strain that is resistant to MetS development (Junková et al, 2021). As they said, hepatic _Acsm3_ was also markedly downregulated in PD strains, which is consistent with our findings.

Medium-chain FAs have been used as an energy source of enteric nutrients because of their rapid metabolism (Schönfeld and Wojtczak, 2016). Medium-chain FAs are directly transported to the liver after passing through the portal vein and are taken up by mitochondria in a carnitine-independent pathway (Kamoshita et al, 2022; Nagao and Yanagita, 2010). Overload of medium-chain FAs, particularly in the liver, promotes FA synthesis, induces hepatic

steatosis, and causes IR (Turner et al, 2009). We found that hepatic medium-chain FAs significantly accumulated in _Acsm3_ knockout mice. Excess FAs were esterified to more glycerol3-phosphate and cholesterol to produce TG or cholesteryl esters, respectively. These neutral lipids can be either stored in cytoplasmic lipid droplets or secreted into the bloodstream as very-low-density lipoprotein particles (Alves-Bezerra and Cohen, 2017). Notably, free FAs could also induce IR through intrahepatocellular activation of several serine/threonine kinases, reduction in tyrosine phosphorylation of the insulin receptor substrate (IRS)-1/2, and impairment of the IRS/PI3K pathway of insulin signaling (Chai et al, 2011).

Lauric acid has been reported to participate in multiple metabolic and cardiovascular diseases (Kamoshita et al, 2022; Miyake et al, 2022; Saraswathi et al, 2020). It should be noted that coconut oil, which contains 50% lauric acid as the primary fatty acid, induces IR in the liver but not in skeletal muscle and adipose tissue (Turner et al, 2009). Lauric acid could also upregulate Hnf4α, which activated Selenop transcription and repaired insulin-induced Akt phosphorylation (Kamoshita et al, 2022). Furthermore, heart failure patients with MetS displayed a higher proportion of lauric acid than heart failure patients without MetS (Lee et al, 2012). In a nonalcoholic fatty liver disease (NAFLD) cohort, higher lauric acid was significantly associated with a high NAFLD activity score in analyses adjusted for the same factors and fibrosis stage (Miyake et al, 2022). In our study, lauric acid was highly accumulated in the livers of _Acsm3_ knockout mice, playing a crucial role in causing the MetS phenotype.

Sufficient evidence indicates an association between p38 MAPK and MetS (Fang et al, 2019; Lan et al, 2022; Wang et al, 2020; Wu et al, 2021; Zhang et al, 2019). P38 MAPK regulates inflammatory activation in metabolic hepatic disease. Macrophage p38α promotes the progression of steatohepatitis by inducing pro-inflammatory cytokine (such as MIP-2α, IL-1β, CXCL10, and IL-6) secretion and M1 polarization (Zhang et al, 2019). P38 MAPK is also involved in arginase-II-mediated endothelial nitric oxide synthase uncoupling in a high-fat diet-induced obesity mouse model, which links obesity-associated IR and type-II diabetes to the increased incidence of cardiovascular disease (Yu et al, 2014). Furthermore, elevated phosphorylation of p38 MAPK is observed in the livers of diabetic mice (Qiao et al, 2006). Its synergistic effects with the phosphorylation of cAMP-response element binding protein enhance hepatic PGC-1α expression to increase hepatic gluconeogenesis (Fang et al, 2019). P38 MAPK activation can also result in

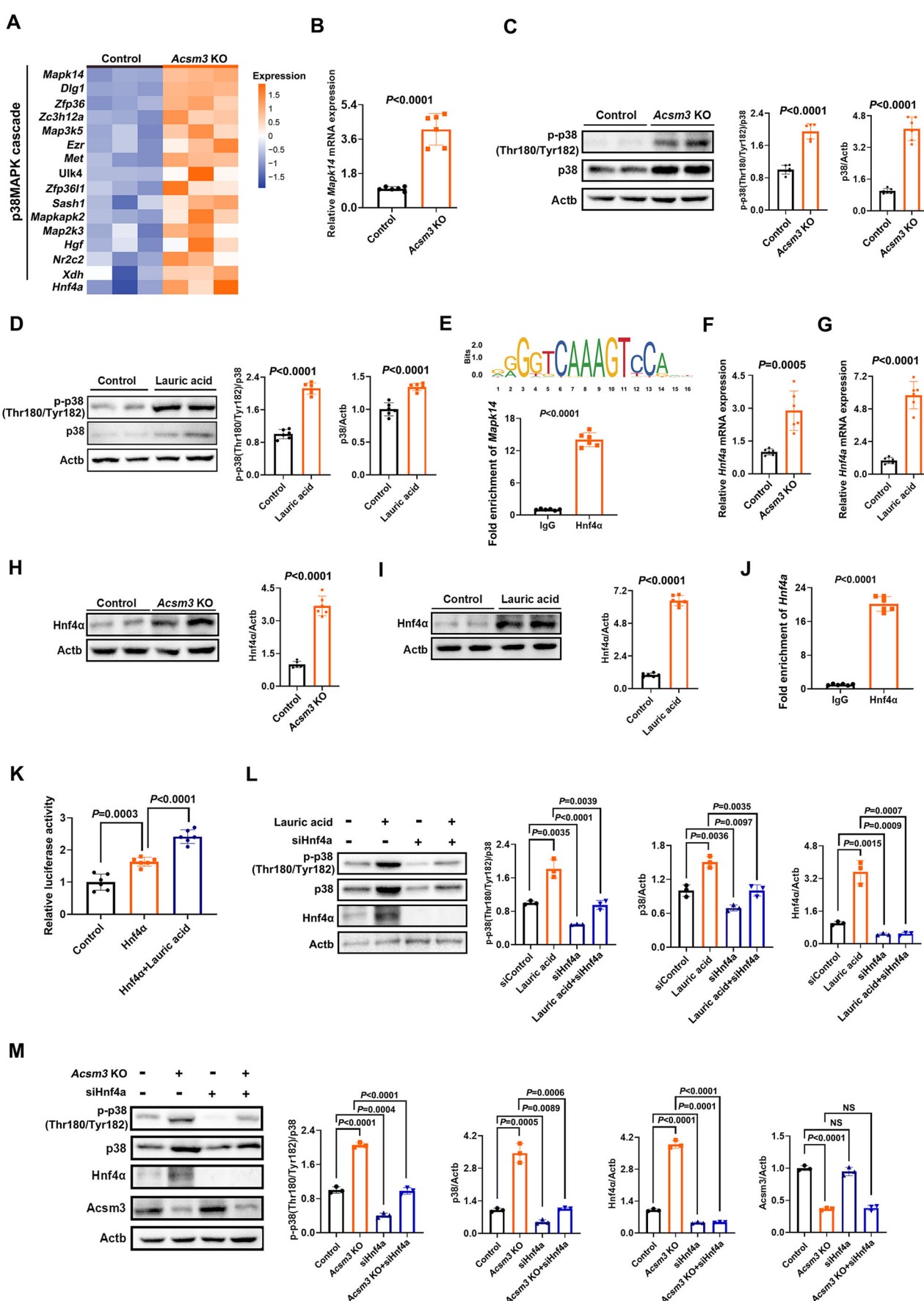

**Figure 5.** The mechanisms of *Acsm3* knockout affected mitochondrial function, thereby causing metabolic syndrome.

All mice in this figure were fed with an FF diet for 8 weeks. (A) Heatmap showing the relative expression of hepatic DEGs (unpaired *t* test, *Acsm3* KO vs control mice) enriched in the p38 MAPK cascade pathway and *Hnf4a* (*n* = 3 biologically independent samples in each group). (B) The relative mRNA expression of *Mapk14* was detected using RT-qPCR (*n* = 6 biologically independent samples in each group). Values were represented as the mean ± SD. Statistics were performed using Student's *t* test. (C) Western blot and quantification of the relative protein expression levels of p-p38 (Thr180/Tyr182) and p38 in the livers of *Acsm3* KO and control mice. Three results for quantification, *n* = 6 biologically independent samples in each group. Values were represented as the mean ± SD. Statistics were performed using Student's *t* test. (D) Western blot and quantification of the relative protein expression of p-p38 (Thr180/Tyr182) and p38 in primary hepatocytes after lauric acid stimulation. Three results were used for quantification, *n* = 6 biologically independent samples in each group. Values were represented as the mean ± SD. Statistics were performed using Student's *t* test. (E) Hnf4α binding site prediction in the *Mapk14* promoter region using JASPAR (http://jaspar.genereg.net/). Chromatin immunoprecipitation (ChIP) was performed to validate that Hnf4α regulated the transcription of *Mapk14* (*n* = 6 biologically independent samples in each group). Values were represented as the mean ± SD. Statistics were performed using Student's *t* test. (F, G) The relative mRNA expression of *Hnf4a* was detected using RT-qPCR assay (*n* = 6 biologically independent samples in each group. Values were represented as the mean ± SD. Statistics were performed using Student's *t* test. (H, I) Western blot and quantification results for the relative protein expressions of Hnf4α. Three results were used for quantification, *n* = 6 biologically independent samples in each group. Values were represented as the mean ± SD. Statistics were performed using Student's *t* test. (J) ChIP was performed to validate that Hnf4α regulated the transcription of *Hnf4a* (*n* = 6 biologically independent samples in each group). Values were represented as the mean ± SD. Statistics were performed using Student's *t* test. (K) The relative luciferase activity showing the binding activation of Hnf4α to its own promoter region after lauric acid stimulation (*n* = 6 biologically independent samples in each group). Values were represented as the mean ± SD. Statistics were performed using Student's *t* test. (L) Western blot and quantification of the relative protein expression of p-p38 (Thr180/Tyr182), p38, and Hnf4α after lauric acid stimulation or Hnf4a knockdown. Three results were used for quantification, *n* = 3 biologically independent samples in each group. Values were represented as the mean ± SD. Statistics were performed using Student's *t* test. (M) Western blot and quantification of the relative protein expression of p-p38 (Thr180/Tyr182), p38, Hnf4α, and Acsm3 after Hnf4a knockdown in *Acsm3* KO or control mouse primary hepatocytes. Three results were used for quantification, *n* = 3 biologically independent samples in each group. Values were represented as the mean ± SD. Statistics were performed using Student's *t* test. Source data are available online for this figure.

mitochondrial dysfunction through increasing ROS levels and oxidative stress (Liu et al, 2022; Manikanta et al, 2020; Wang et al, 1992). In our study, the activation of p38 MAPK signal transduction explains the abnormal phenotype of glucose/lipid metabolism and mitochondrial dysfunction caused by *Acsm3* knockout.

There are some limitations in this study. The major limitation is that mouse data were only collected in males when MetS equally affects females. Some evidence suggests sex differences in FA metabolism (Adler-Wailes et al, 2013; Blaak, 2001; Decsi and Kennedy, 2011; Jensen, 1995). Adult women release more free FA relative to resting energy expenditure and have greater free FA clearance rates than men (Adler-Wailes et al, 2013). In adolescents, independent of the effects of resting energy expenditure and fat mass, free FA kinetics differ significantly in obese adolescent girls and boys, with greater free FA flux among girls(Adler-Wailes et al, 2013). Furthermore, sex hormones may influence the enzymatic synthesis of long-chain polyunsaturated FAs, which may lead to sex-specific differences in long-chain polyunsaturated FA status (Decsi and Kennedy, 2011). A review of the literature generally suggested that there was a higher contribution of arachidonic acid and docosahexaenoic acid in blood lipids in women than in men; however, sex-specific differences were not seen in every study (Blaak, 2001; Decsi and Kennedy, 2011; Jensen, 1995). Further experimental studies are needed to elucidate the effects of sex differences on Acsm3 participation in FA metabolism. Another limitation is that although lauric acid is the most specific medium-chain FA catalyzed by Acsm3, multiple other FAs also accumulated after *Acsm3* knockout. We cannot determine whether one of the other FAs would have the same or even more severe effect as lauric acid (such as activating p38 MAPK), which should be explored in further studies.

In summary, we clarified the mechanism by which Acsm3 is involved in MetS. Acsm3 deletion caused lauric acid accumulation, further promoting *Hnf4a* and *Mapk14* transcriptions to activate p38 MAPK signaling, which impaired mitochondrial function and contributed to abnormal glucose and lipid metabolism. The p38

MAPK inhibitor adezmapimod could rescue the metabolic disorders caused by Acsm3 deficiency. Taken together, these findings provided novel insights into the roles of ACSM3 in MetS and hinted at new therapeutic strategies.

# Methods

## The first cohort for genome-wide screening

The study population was from Rizhao City, in the northern region of China from 2009 to 2010 which has been described previously (Li et al, 2014). The following strict inclusion criteria were used for both MetS patients and controls: (1) Chinese Han people; (2) Male; (3) aged 48–66; and (4) no evidence of thyroid disease, hematological diseases, peptic ulcers, liver or kidney dysfunctions, infections, autoimmune diseases, or tumors. The more strict criteria for MetS were used as BMI ≥ 28.0 or BMI > 27.0 & waist circumference >101 cm, plus three or more of the following: (1) elevated triglycerides (drug treatment for elevated triglycerides is an alternate indicator) ≥1.7 mmol/L; (2) reduced HDL-C (drug treatment for reduced HDL-C is an alternate indicator) <1.0 mmol/L; (3) elevated blood pressure (antihypertensive drug treatment in a patient with a history of hypertension is an alternate indicator) with current or previous SBP ≥ 160 mmHg and DBP ≥ 100 mmHg; (4) elevated fasting glucose (drug treatment of elevated glucose is an alternate indicator) ≥6.1 mmol/L. The criteria for controls included 19.5 < BMI < 24.0, SBP < 120 mmHg & DBP < 80 mmHg, TG < 1.7 mmol/L, HDL ≥ 1.0 mmol/L, fasting glucose <6.1 mmol/L, and no history of hypertension, hyperlipidemia, diabetes, cardiovascular disease, or stroke. Females in this cohort were around menopause, and the sex hormones declined sharply during this period, which would affect metabolism. To discriminate against such bias, we excluded females in our first rigorous cohort. Written informed consent was obtained from all subjects (Cui et al, 2021). The ethics committee of Rizhao Port Hospital and Fuwai Hospital approved the study.

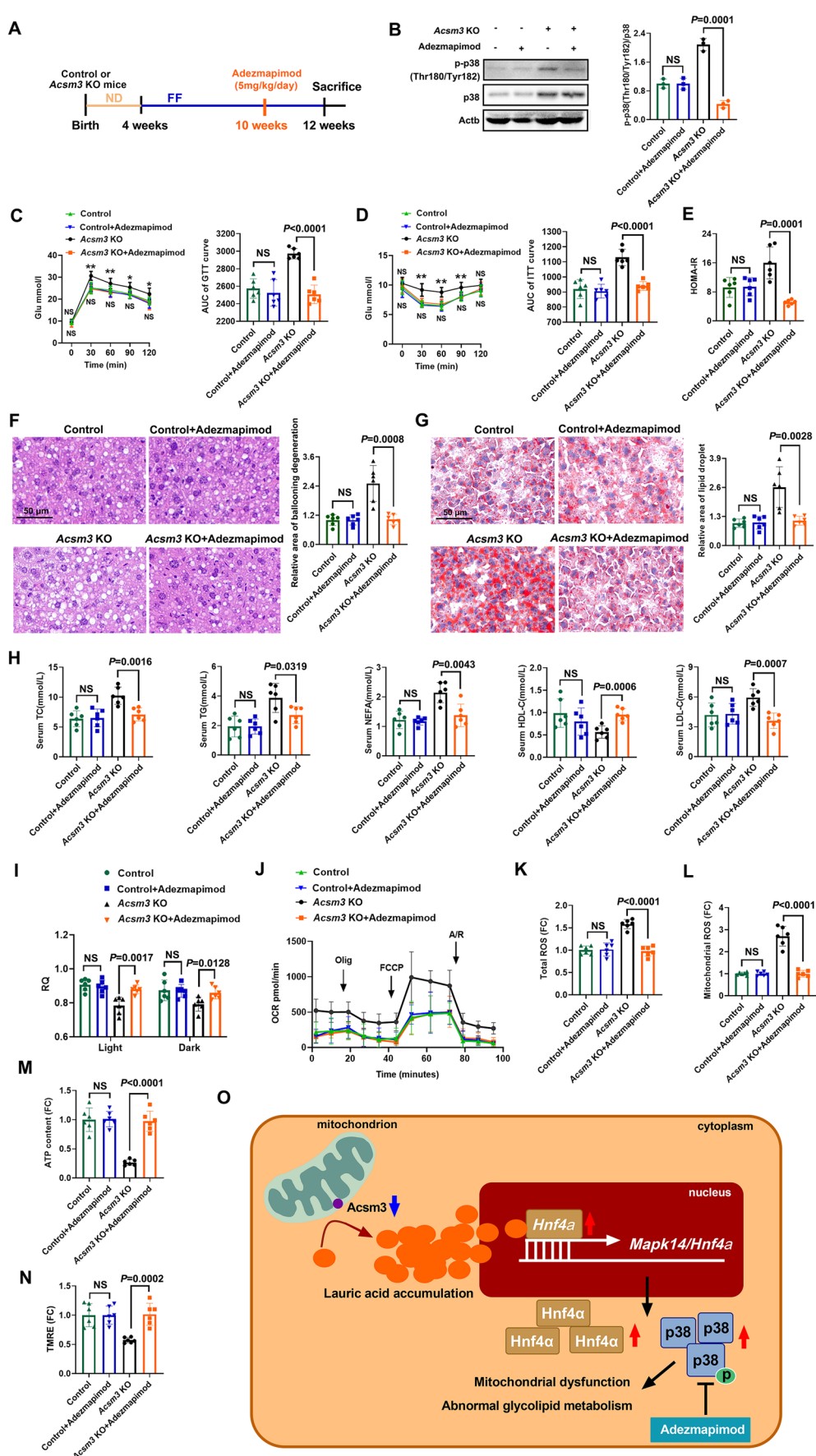

**Figure 6. Adezmapimod rescued the phenotype caused by Acsm3 deletion.**

All mice in this figure were fed with an FF diet for 8 weeks. (A) Timeline of adezmapimod (5 mg/kg/day) treatment and FF diet in *Acsm3* KO and control mice. (B) Western blot and quantification of the relative protein expression of p-p38 (Thr180/Tyr182) and p38 in the livers of *Acsm3* KO and control mice treated with adezmapimod or not. Three results were used for quantification, $n = 3$ biologically independent samples in each group. Values were represented as the mean ± SD. Statistics were performed using Student's *t* test. (C, D) GTT, ITT, and their respective areas under the AUC ($n = 6$ biologically independent samples in each group). Values were represented as the mean ± SD. Statistics were performed using Student's *t* test. *P* values were denoted by asterisks: *$P < 0.05$, **$P < 0.01$. The values above the curve represent the statistical values of the *Acsm3* KO group vs *Acsm3* KO with adezmapimod treatment group, and the values below the curve represent the statistical values of the control group vs control group with adezmapimod treatment. (E) The HOMA-IR indexes ($n = 6$ biologically independent samples in each group). HOMA-IR = fasting Glu (mmol/L) × fasting Ins (μU/mL)/22.5. Values were represented as the mean ± SD. Statistics were performed using Student's *t* test. (F, G) Representative images of H&E and oil red O staining ($n = 6$ biologically independent samples in each group). Scale bar, 50 μm. The quantification of the relative area of ballooning degeneration or lipid droplets was based on NIH ImageJ software. Values were represented as the mean ± SD. Statistics were performed using Student's *t* test. (H) The contents (mmol/L) of serum TC, TG, NEFA, HDL-C, and LDL-C ($n = 6$ biologically independent samples in each group). Values were represented as the mean ± SD. Statistics were performed using Student's *t* test. (I) RQ in light and dark, respectively ($n = 6$ biologically independent samples in each group). Values were represented as the mean ± SD. Statistics were performed using Student's *t* test. (J) OCR analyses of primary hepatocytes. Olig, FCCP, or A/R treatments were given at the indicated time points ($n = 6$ biologically independent samples in each group). Values were represented as the mean ± SD. Statistics were performed using Student's *t* test. (K, L) Relative levels of total ROS and mitochondrial ROS in the mouse liver. The data show the MFIs ($n = 6$ biologically independent samples in each group). Values were represented as the mean ± SD. Statistics were performed using Student's *t* test. (M) The relative contents of ATP in primary hepatocytes ($n = 6$ biologically independent samples in each group). Values were represented as the mean ± SD. Statistics were performed using Student's *t* test. (N) Relative mitochondrial membrane potentials of the primary hepatocytes were detected using a TMRE probe ($n = 6$ biologically independent samples in each group). Data show the MFI. Values were represented as the mean ± SD. Statistics were performed using Student's *t* test. (O) Schematic illustration showing the mechanism of Acsm3 in metabolic syndrome. Acsm3 deficiency resulted in lauric acid accumulation, thereby increasing the transcription levels of *Hnf4a* and *Mapk14*, thus activating the p38 MAPK signaling pathway and leading to mitochondrial dysfunction and abnormal glucose and lipid metabolism. The p38 MAPK inhibitor adezmapimod effectively rescued mitochondrial dysregulation and improved the MetS phenotype caused by Acsm3 deletion in mice. Source data are available online for this figure.

## The second cohort for validation

The second validation cohort was selected from a cohort established for a resistant hypertension study in China (Wu et al, 2018; Xiao et al, 2022). The following inclusion criteria were used for both MetS and controls: (1) Chinese Han people; (2) aged 60–75 years; (3) no evidence of thyroid disease, hematological diseases, peptic ulcers, liver or kidney dysfunctions, infections, autoimmune diseases, or tumors. The criteria for MetS were used as waist circumference ≥ 95 cm along with the presence of two or more of the following: elevated triglycerides (or drug treatment) ≥1.7 mmol/L, reduced HDL-C (or drug treatment) <1.0 mmol/L, elevated blood pressure (or drug treatment) with systolic ≥140 and/or diastolic ≥90 mmHg, and elevated fasting glucose (or drug treatment) ≥6.1 mmol/L. The inclusion criteria for the controls were the same as those for the first cohort (Cui et al, 2021). The ethics committee of Rizhao Port Hospital and Fuwai Hospital approved the study.

## Microarray gene expression analysis

Gene expression profiling analysis was performed by Shanghai Biotechnology Corporation (Shanghai, China). The Affymetrix human gene 2.0 ST profile chip was used to detect the whole transcript level. RNA from samples was isolated according to the manufacturer's protocol (Qiagen). RNA qualities were assessed using an Agilent 2100 bioanalyzer and the RNA 6000 Nano Chip (Agilent Technologies), and RNA quantities were determined using a Nanodrop-1000 Spectrophotometer (Thermo Scientific, Hudson, NH). Briefly, 300 ng of total RNA per sample was converted to double-strand cDNA using the procedure recommended by Affymetrix (http://www.affymetrix.com). Using a random hexamer incorporating a T7 promoter, amplified RNA (cRNA) was generated from the double-stranded cDNA template by IVT (in-vitro transcription) and then purified using the Affymetrix sample cleanup module. cDNA was regenerated by random-primed reverse

transcription using a dNTP mix containing dUTP and then fragmented using UDG and APE1 restriction endonucleases and end-labeled with biotinylated dideoxynucleoside using the terminal transferase reaction. Fragmented end-labeled cDNA was hybridized to GeneChip® Human Gene 2.0 ST arrays for 17 h at 45 °C and 60 rpm as described in the Gene Chip Whole Transcript (WT) Sense Target Labeling Assay Manual (Affymetrix). After hybridization, chips were stained and washed in a Genechip Fluidics Station 450 (Affymetrix) and scanned using a Genechip Array scanner 3000 7 G (Affymetrix). Expression intensities were extracted from scanned images using Affymetrix Command Console software version 1.1 and stored as CEL files. Raw data were normalized by Expression Console provided by Affymetrix (http://www.affymetrix.com).

## Animal study

All animal use and welfare adhered to the National Institutes of Health's Guide for the Care and Use of Laboratory Animals following a protocol reviewed and approved by the State Key Laboratory of Cardiovascular Disease, National Center for Cardiovascular Diseases (Beijing, China; permit number: 0000869). All animals were housed under a 12 h light/dark cycle at a temperature of 24 °C and relative humidity of (56 ± 10)%, with free access to water and a normal diet (ND).

*Acsm3* knockout mice (C57BL/6 J background) were constructed at the Nanjing Biomedical Research Institute of Nanjing University. The targeting strategy and expression identification of knockout mice are shown in Fig. 2A,B. Male mice are more sensitive to diet-induced MetS than female mice, and thus male mice were used in this study (Kleinert et al, 2018; Leonardi et al, 2020).

Acsm3 liver-specific knockdown mice were generated via adenovirus injection. The adeno-associated virus 8 (AAV8) system carrying shRNA against scramble (as a negative control) or Acsm3 (designed and synthesized by GeneChem, Shanghai, China) was

transduced into mice at a dose of $1 \times 10^{12}$ vg through tail-vein injection (Chen et al, 2019).

To induce MetS, at the age of 4 weeks, C57BL/6 J mice were then fed a high-fat and high-fructose (FF) diet (40 kcal% fat, 20 kcal% fructose, and 2% cholesterol, D09100301, Research Diets, Inc., NJ, USA) for 8 weeks (Chou et al, 2018; Zhang et al, 2020).

Adezmapimod was dissolved in Dimethyl sulfoxide (DMSO) and used to rescue the phenotype induced by *Acsm3* knockout. Specifically, adezmapimod (5 mg/kg/day) or the same dose of DMSO was daily intraperitoneal injected into mice for sixteen consecutive days (Lali et al, 2000). After the last injection, the mice were sacrificed and used for subsequent analysis.

## Isolation and culture of primary hepatocytes

Primary hepatocytes were isolated from C57BL/6 J mice by a two-step collagenase method as described by Li et al (Li et al, 2010). Briefly, in the first step, a calcium-free medium containing a calcium chelator was perfused through the liver. Removal of calcium ions (by EDTA, EGTA, or citrate) from epithelial cells results in the rapid destruction of intercellular junctions with the consequence that the cell–cell contacts are lost. The second step was the introduction of the enzyme collagenase into the liver lobes disrupting the supporting extracellular matrix.

Freshly isolated hepatocytes were seeded in DMEM (Dulbecco's modification of Eagle's medium, Gibco, USA) containing 10% FBS (fetal bovine serum, Gibco, USA) and 5 ng/ml HGF (hepatocyte growth factor, Thermo Fisher Scientific, USA). The cells were maintained at 37 °C in 5% $CO_2$/95% air, and the media were changed every second day.

## RNA sequencing of liver tissues

Transcriptome sequencing and analysis were conducted by Shanghai Biotechnology Corporation (Shanghai, China). Total RNA was extracted using the MJzol Animal RNA Isolation Kit (Majorivd) and purified using the RNAClean XP Kit (Beckman Coulter) and RNase-Free DNase Set (QIAGEN) according to standard operating procedures provided by the manufacturer. The mRNA was enriched using VAHTS® mRNA Capture Beads (Human/Mouse/Rat). The mRNA sequencing libraries were constructed using the VAHTS Universal V6 RNA-seq Library Prep Kit for Illumina® (Vazyme) according to standard operating procedures provided by the manufacturer.

## Glucose tolerance test

Mice were fasted overnight and intraperitoneally injected with glucose (2 g/kg body weight) for the glucose tolerance test (GTT). Blood glucose levels were measured using blood glucose meters (Roche, Switzerland) at 0, 30, 60, 90, and 120 min after injection. Blinding of experimental conditions was performed for GTT.

## Insulin tolerance test

Mice were fasted for 6 h and intraperitoneally injected with human insulin (1 unit/kg body weight) for an insulin tolerance test (ITT). Blood glucose levels were measured using blood glucose meters

(Roche, Switzerland) at 0, 30, 60, 90, and 120 min after injection. Blinding of experimental conditions was performed for ITT.

## Insulin measurement

Serum insulin was detected by an ELISA kit for mouse insulin (Abcam, UK).

## Histological analysis

Mouse livers were fixed, dehydrated, and embedded in paraffin. Hematoxylin and eosin (H&E) staining was performed to visualize hepatic pathological changes. Oil Red O staining of frozen liver sections was used to visualize lipid droplet accumulation. The quantification of the relative area of ballooning degeneration or lipid droplets was based on NIH ImageJ software (Kleiner et al, 2005; Li et al, 2022).

## Transmission electron microscopy for hepatic mitochondria morphology

Liver tissues were sliced into 1-mm cubes, placed in a fixative buffer (2.5% glutaraldehyde) overnight (2 h at room temperature and 12 h at 4 °C), postfixed in 1% osmium tetroxide in 0.1 mol/L sodium cacodylate buffer for 1 h on ice, and stained en bloc with 2% uranyl acetate for 1 h on ice. The stained tissues were dehydrated in ethanol (20–100%) and embedded with Durcupan (Sigma-Aldrich, USA). Ultrathin (50–60 nm) sections were post-stained with uranyl acetate and lead stain. Samples were viewed and photographed using a PHILIPSCM120-TEM (Philip, Netherlands). Mitochondrial areas and lengths were quantified using NIH ImageJ software.

## Analysis of tissue lipids

Hepatic lipids were extracted using the Folch method (Jiménez et al, 2012) and quantified using ultraperformance liquid chromatography/mass spectrometry and direct-infusion mass spectrometry (Zhao et al, 2014).

## Biochemical parameters

Serum total cholesterol (TC), triglyceride (TG), nonesterified fatty acid (NEFA), high-density lipoprotein cholesterol (HDL-C), low-density lipoprotein cholesterol (LDL-C) and hepatic TC, TG, and NEFA were detected using kits from Nanjing Jiancheng Bioengineering Institute.

Serum alanine aminotransferase (ALT) and aspartate aminotransferase (AST) were analyzed to evaluate liver function using commercial kits (Beijing Ruizheng Shanda Biological Engineering Technology Co. Ltd, China) with an automatic biochemical analyzer (Chen et al, 2020).

Total ROS levels (Solarbio, China), mitochondrial ROS levels (Bestbio, China), ATP contents (Beyotime, China), and mitochondrial membrane potential (Beyotime, China) in livers were detected according to the manufacturer's instructions. Intensity readings for the above assays were taken with an automatic microplate reader (Tecan, Switzerland).

## Respiratory measurements

The mice were individually housed in metabolic cages (Oxylet, PanLab, Spain) to measure $O_2$ consumption ($VO_2$) and $CO_2$ production ($VCO_2$). Spontaneous activities were monitored by activity sensors at the same time. The respiratory quotient ($RQ = VCO_2/VO_2$) was calculated using Metabolism software.

## Oxygen consumption rate

The oxygen consumption rate (OCR) assay was performed using an XF24 Extracellular Flux Analyzer (Agilent, USA) following the manufacturer's instructions. Hepatocytes were seeded into 24-well culture plates ($2 \times 10^4$ cells per well). Oligomycin (Olig), trifluoromethoxy carbonyl cyanide phenylhydrazone (FCCP), and antimycin/rotenone (A/R) were added to inhibit ATP synthase, uncouple oxygen consumption from ATP production, and inhibit complexes I and III, respectively.

## Chromatin immunoprecipitation

A chromatin immunoprecipitation (ChIP) assay was performed using the ChIP-IT Express enzymatic kit (Active Motif, USA) according to the manufacturer's instructions. Cell chromatin was fixed, fragmented, and then incubated with protein G-coated magnetic beads and antibodies overnight at 4 °C. After elution of chromatin and reverse cross-linking, DNA was applied for PCR analysis. The primer sequences for the ChIP assay are shown in Table EV1.

## Dual-luciferase reporter assay

The mouse *Hnf4a* promoter sequences were inserted into the NheI/HindIII site of the luciferase pGL3-Basic vector (Promega, USA). The primary hepatocytes were divided into three groups: the control group (transfected with reporter vector pGL3-*Hnf4a* promoter and control plasmid), Hnf4α group (transfected with reporter vector pGL3-*Hnf4a* promoter and *Hnf4a* plasmid), and Hnf4α + lauric acid group (transfected with reporter vector pGL3-*Hnf4a* promoter and *Hnf4a* plasmid, then stimulated with lauric acid). All groups were cotransfected with target vectors and pRL-TK (Promega, USA). After 24 h of incubation, the cells were treated with DMSO or lauric acid (0.5 mmol/L) for 24 h. Next, the dual-luciferase reporter assay was performed according to the manufacturer's instructions (Promega, USA).

## Real-time quantitative PCR

Total RNA was isolated using the Trizol reagent (Invitrogen, USA). 1 μg of total RNA was converted to cDNA, and real-time quantitative PCR (RT-qPCR) (Promega, USA) was performed according to standard protocols. Each sample was run in triplicate to ensure quantitative accuracy, and the threshold cycle numbers (Ct) were averaged. The results were calculated using the $2^{-\Delta\Delta Ct}$ method (Ren et al, 2023). The RT-qPCR primer sequences are shown in Table EV1.

## Western blot analysis

Protein samples were subjected to SDS-PAGE, transferred to polyvinylidene fluoride (PVDF) membranes, and blocked with fat-free milk. The membranes were incubated overnight at 4 °C with primary antibodies against Acsm3 (1:500, Santa, sc-377173), p-p38 (Thr180/Tyr182) (1:1000, CST, 4511), p38 (1:1000, CST, 9212), Hnf4α (1:500, Santa, sc-374229), and Actb (1:5000, Proteintech, 20536-1-AP). The blots were rinsed with Tris-buffered saline Tween (TBST) and incubated with a secondary antibody for 1 h at room temperature. ECL Supersensitive luminescent solutions were added to the membranes and developed using a gel imager (BIO-RAD, USA). NIH ImageJ software was used to analyze the gray value of proteins (Hong et al, 2021).

## Statistical analysis

Differentially expressed genes (DEGs) were identified with unpaired *t* tests. For pathway enrichment analysis, *P* values were computed with the hypergeometric test and adjusted in a Benjamini–Hochberg procedure for multiple hypothesis correction. All experimental data are presented as the means ± standard deviations (SDs) and were analyzed by Student's *t* test for *P* value determination. All statistical analyses were performed with R (https://www.r-project.org/), python (https://www.python.org/), or PRISM (GraphPad Software Inc) (Yan et al, 2021).

# Data availability

The data supporting the findings of this study are available in the manuscript and its supplementary information. The RNA sequencing data for Figs. 4A and 5A are available at GSE247290. The source data for Fig. 3J,K are provided in Dataset EV1. All other data for figures in the manuscript are provided in the source data file.

# Peer review information

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

## Acknowledgements

We thank all the patients and their families for participating in our study, and for offering all information, and data. This work was supported by the Chinese Academy of Medical Sciences with Innovation Fund for Medical Sciences grant (2021-I2M-1-016), the National Natural Science Foundation (81970430 and 81770424), the Longevity Pilot Project grant (2019-RC-HL-002), and National Key R&D Program of China with grant 2017YFC0909400.

## Author contributions

**Xiao Xiao**: Software; Formal analysis; Supervision; Validation; Investigation; Visualization; Methodology; Writing—original draft. **Ruofei Li**: Software; Formal analysis; Supervision; Validation; Investigation; Visualization; Methodology; Writing—original draft; Project administration; Writing—review and editing. **Bing Cui**: Software; Formal analysis; Investigation; Visualization; Methodology; Writing—original draft. **Cheng Lv**: Software; Investigation; Methodology. **Yu Zhang**: Data curation. **Jun Zheng**: Data curation. **Rutai Hui**: Data curation. **Yibo Wang**: Conceptualization; Resources; Data curation; Software; Formal analysis; Supervision; Funding acquisition; Methodology; Project administration; Writing—review and editing.

## Disclosure and competing interests statement

The authors declare no competing interests.

# Expanded View Figures

**A**

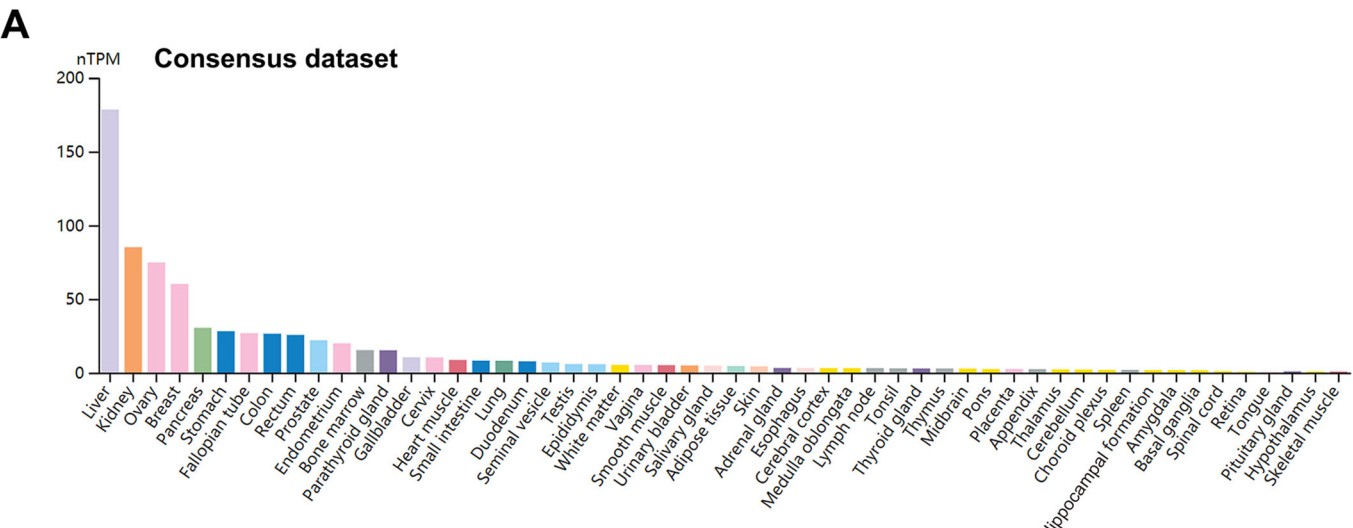

**B**

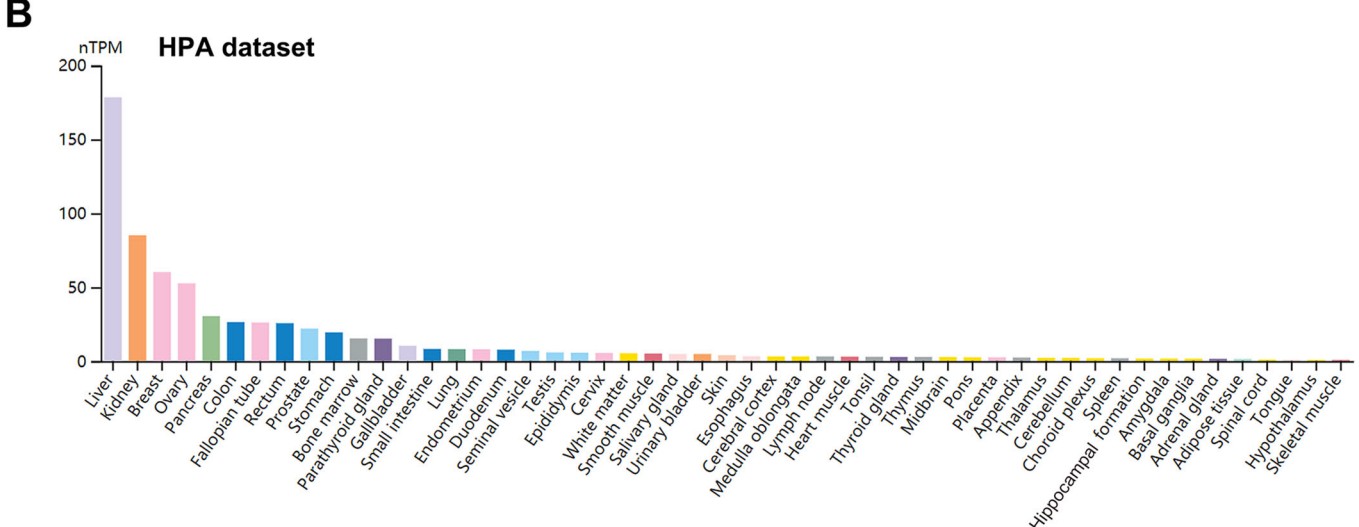

**Figure EV1. Expression distributions of Acsm3.**

(A, B) Consensus dataset and HPA dataset showing the expression of Acsm3 in different tissues in the human protein atlas database (https://www.proteinatlas.org/).

**A**

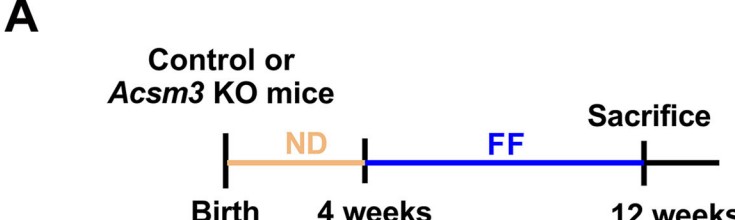

**B**

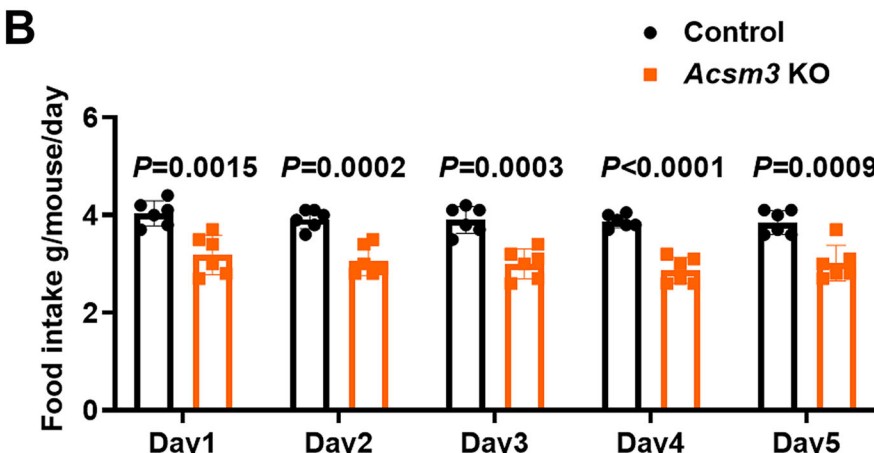

**C**            **D**

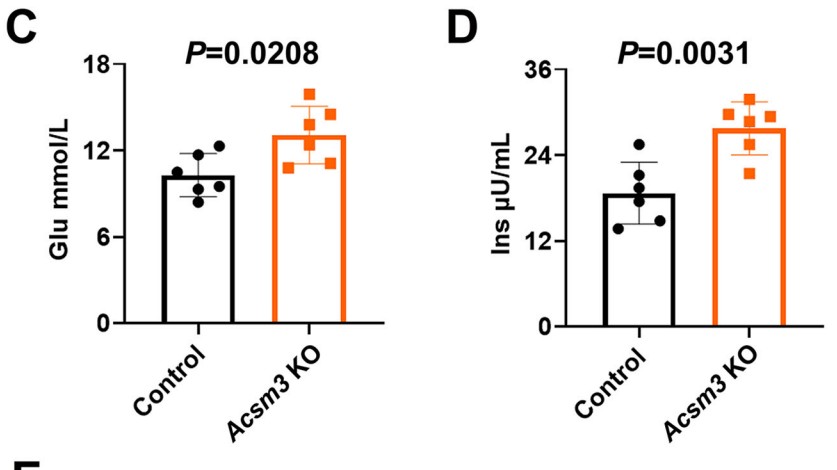

**E**

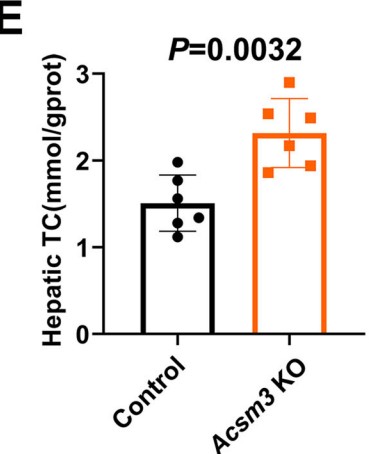
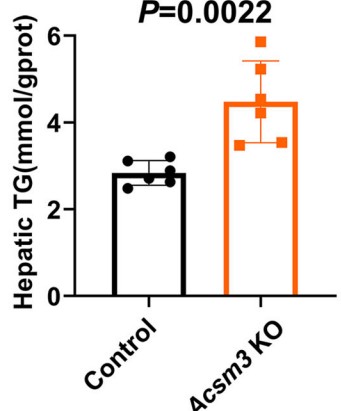
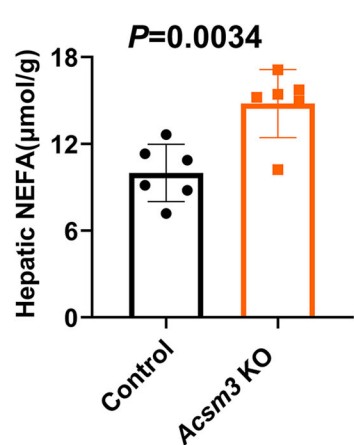

◀   **Figure EV2.  *Acsm3* knockout mice under high-fat and high-fructose diet.**

(A) Timeline of *Acsm3* KO and control mice fed with an FF diet. (B) The average daily food intake of *Acsm3* KO and control mice for 5 consecutive days ($n = 6$ biologically independent samples in each group). Values were represented as the mean ± SD. Statistics were performed using Student's *t* test. (C, D) The fasting Glu (mmol/L), fasting Ins (μU/mL) of *Acsm3* KO and control mice. (E) The contents of hepatic TC (mmol/gprot), TG (mmol/gprot), and NEFA (μmol/g) in *Acsm3* KO and control mice ($n = 6$ biologically independent samples in each group). Values were represented as the mean ± SD. Statistics were performed using Student's *t* test.

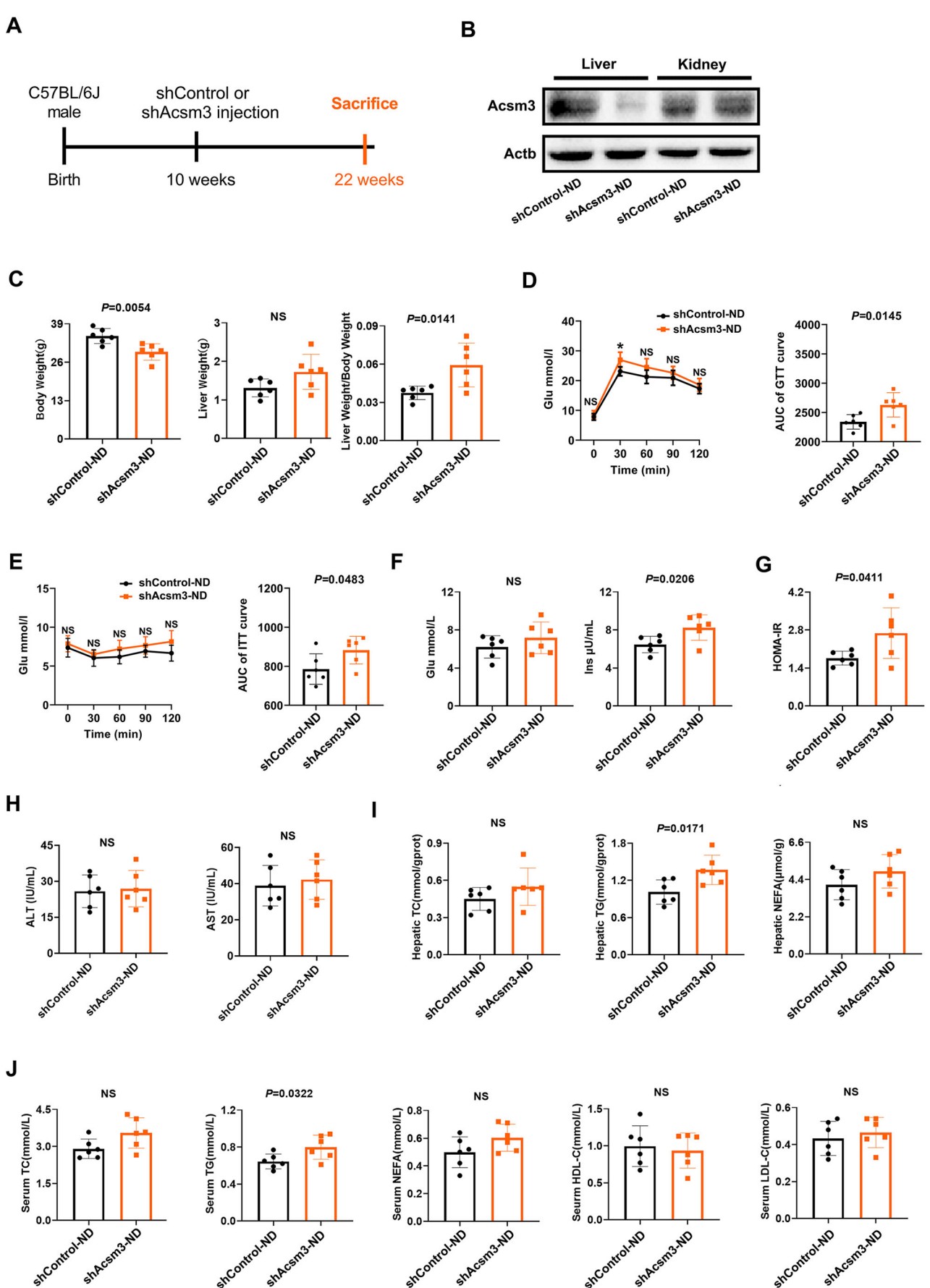

◄ **Figure EV3. Acsm3 liver-specific knocked down mice under normal diet.**

(**A**) Timeline of shAcsm3 mice (Adeno-associated virus 8 (AAV8) injection induced liver-specific knockdown of Acsm3 in mice) and shControl mice under ND. (**B**) Western blot showing the relative expression of Acsm3 in the liver and kidney of shAcsm3 and shControl mice. (**C**) The body weights (g), liver weights (g), and liver weight/body weight of shAcsm3 and shControl mice ($n = 6$ biologically independent samples in each group). Values were represented as the mean ± SD. Statistics were performed using Student's *t* test. (**D, E**) GTT, ITT, and respective AUC of shAcsm3 and shControl mice ($n = 6$ biologically independent samples in each group). Values were represented as the mean ± SD. Statistics were performed using Student's *t* test. *P* values were denoted by asterisks: *$P < 0.05$. (**F, G**) The fasting Glu (mmol/L), fasting Ins (μU/mL), and HOMA-IR indexes of shAcsm3 and shControl mice ($n = 6$ biologically independent samples in each group). HOMA-IR = fasting Glu (mmol/L) × fasting Ins (μU/mL)/22.5. Values were represented as the mean ± SD. Statistics were performed using Student's *t* test. (**H**) The contents (IU/mL) of serum ALT and AST in shAcsm3 and shControl mice ($n = 6$ biologically independent samples in each group). Values were represented as the mean ± SD. Statistics were performed using Student's *t* test. (**I**) The contents of hepatic TC (mmol/gprot), TG ((mmol/gprot)), and NEFA (μmol/g) in shAcsm3 and shControl mice ($n = 6$ biologically independent samples in each group). Values were represented as the mean ± SD. Statistics were performed using Student's *t* test. (**J**) The contents (mmol/L) of serum TC, TG, NEFA, HDL-C, and LDL-C in shAcsm3 and shControl mice ($n = 6$ biologically independent samples in each group). Values were represented as the mean ± SD. Statistics were performed using Student's *t* test.

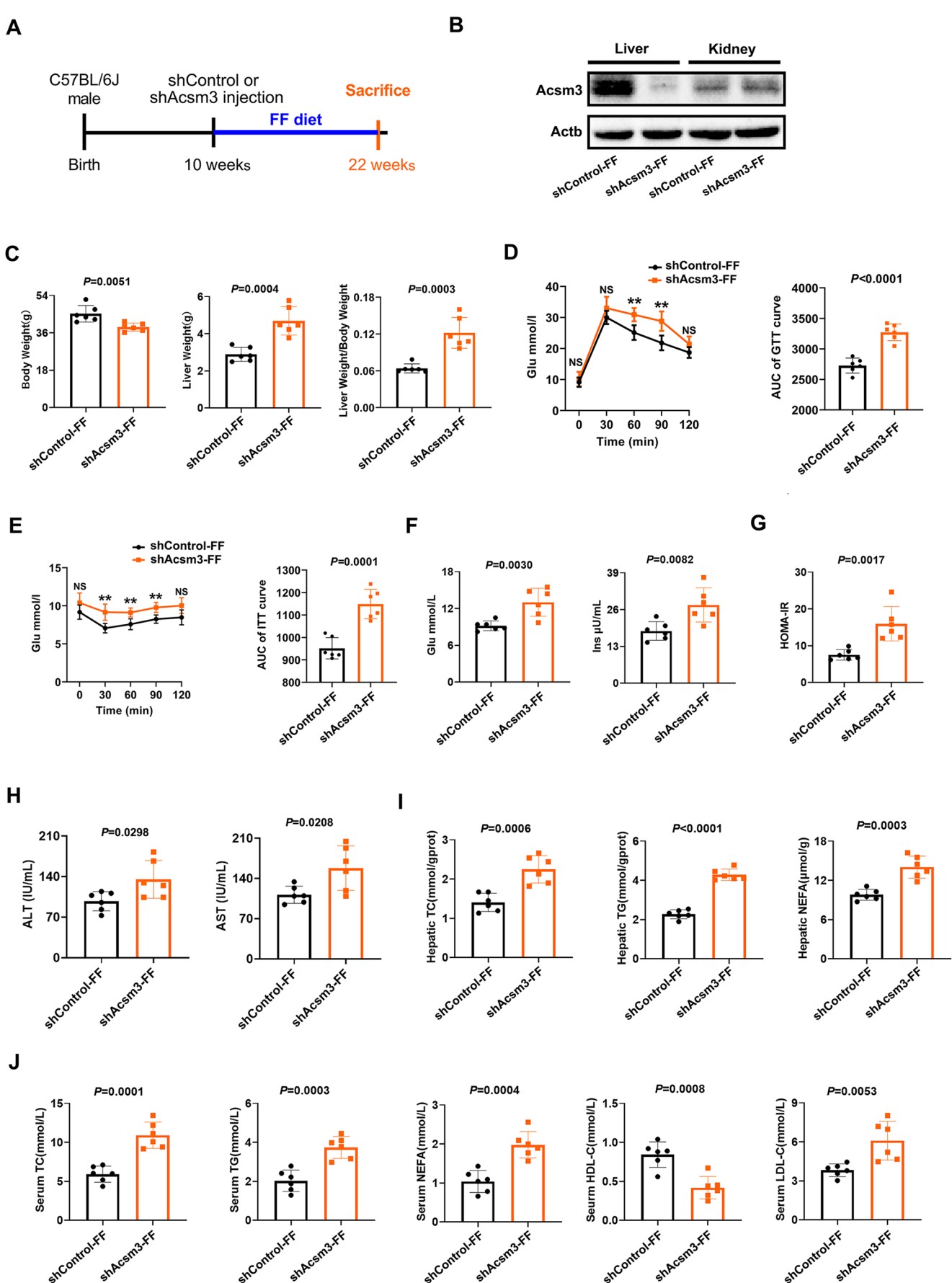

◀ **Figure EV4. Acsm3 liver-specific knocked down mice under FF diet.**

(A) Timeline of shAcsm3 mice and shControl mice fed with an FF diet. (B) Western blot showing the relative expression of Acsm3 in the liver and kidney of shAcsm3 and shControl mice. (C) The body weights (g), liver weights (g), and liver weight/body weight of shAcsm3 and shControl mice ($n = 6$ biologically independent samples in each group). Values were represented as the mean ± SD. Statistics were performed using Student's $t$ test. (D, E) GTT, ITT, and respective AUC of shAcsm3 and shControl mice ($n = 6$ biologically independent samples in each group). Values were represented as the mean ± SD. Statistics were performed using Student's $t$ test. $P$ values were denoted by asterisks: **$P < 0.01$. (F, G) The fasting Glu (mmol/L), fasting Ins (μU/mL), and HOMA-IR indexes of shAcsm3 and shControl mice ($n = 6$ biologically independent samples in each group). HOMA-IR = fasting Glu (mmol/L) × fasting Ins (μU/mL)/22.5. Values were represented as the mean ± SD. Statistics were performed using Student's $t$ test. (H) The contents (IU/mL) of serum ALT and AST in shAcsm3 and shControl mice ($n = 6$ biologically independent samples in each group). Values were represented as the mean ± SD. Statistics were performed using Student's $t$ test. (I) The contents of hepatic TC (mmol/gprot), TG ((mmol/gprot)), and NEFA (μmol/g) in shAcsm3 and shControl mice ($n = 6$ biologically independent samples in each group). Values were represented as the mean ± SD. Statistics were performed using Student's $t$ test. (J) The contents (mmol/L) of serum TC, TG, NEFA, HDL-C, and LDL-C in shAcsm3 and shControl mice ($n = 6$ biologically independent samples in each group). Values were represented as the mean ± SD. Statistics were performed using Student's $t$ test.

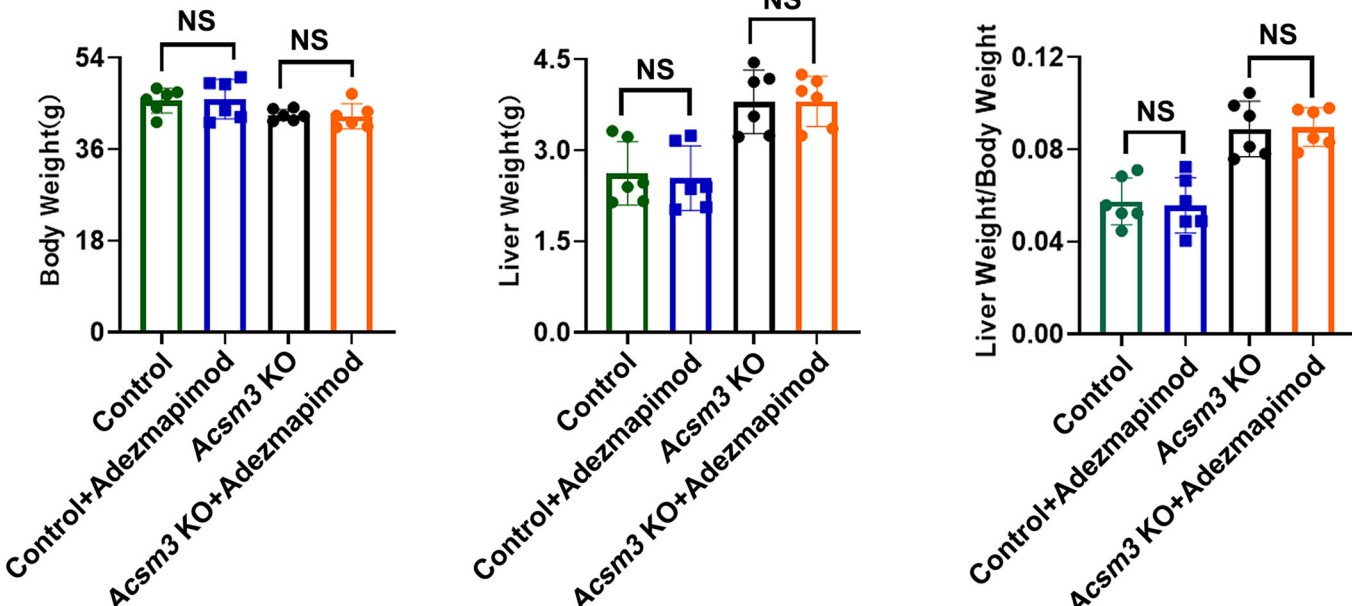

**Figure EV5.   The body weights (g), liver weights (g), and liver weight/body weight of *Acsm3* KO and control mice treated with adezmapimod or not.**

Values were represented as the mean ± SD (*n* = 6 biologically independent samples in each group). Statistics were performed using Student's *t* test.

