## [Peer Review File · The EMBO Journal]

Liver ACSM3 deficiency mediates metabolic syndrome via a lauric acid-HNF4 α -p38 MAPK axis

Xiao Xiao, Ruofei Li, Bing Cui, Cheng Lv, Yu Zhang, Jun Zheng, Rutai Hui, and Yibo Wang
DOI: [10.15252/emboj.2023114416](https://doi.org/10.15252/emboj.2023114416)

Corresponding author: Yibo Wang (wangyibo@fuwaihospital.org)

Review Timeline:

Submission Date:	2nd May 23
Editorial Decision:	18th Aug 23
Revision Received:	12th Nov 23
Editorial Decision:	26th Nov 23
Revision Received:	27th Nov 23
Accepted:	30th Nov 23

Editor: Daniel Klimmeck

Transaction Report:

Dear Dr Wang,

Thank you for the submission of your manuscript (EMBOJ-2023-114416) to The EMBO Journal, as well as for your kind patience with our response at this time of the year. Your study has been sent to two reviewers with expertise in metabolism and lipid biology and we have received feedback from both of them, which I enclose below.

As you will see, the referees acknowledge the potential interest and novelty of your results, although they also express several important issues that will have to be conclusively addressed before they can be supportive of publication of your manuscript in The EMBO Journal. In more detail, the experts point to insufficient integration of the human MetS cohort data and detailing a rationale for the focus on ACSM3, as well as its exact role and causalities in the development of the disease. Also, the experts request consolidation of your results by complementary analysis of lipids (ref#1) and liver parameters (ref#2). Further, the reviewers raise a number of points related to additional controls required, improved methods annotation and data processing and overall discussion of the findings, that would need to be conclusively addressed to achieve the level of robustness and clarity needed for The EMBO Journal.

I judge the comments of the referees to be generally reasonable and given their overall interest, we are in principle happy to invite you to revise your manuscript experimentally to address the referees' comments.

As you may have seen on our web page, we generally allow three months as standard revision time. As a matter of policy, competing manuscripts published during this period will not negatively impact on our assessment of the conceptual advance presented by your study. However, we request that you contact the editor as soon as possible upon publication of any related work, to discuss how to proceed. Should you foresee a problem in meeting this three-month deadline, please let us know in advance and we may be able to grant an extension.

When submitting your revised manuscript, please carefully review the instructions below.

Thank you for the opportunity to consider your work for publication.
I look forward to your revision.

Kind regards,

Daniel Klimmeck

Daniel Klimmeck, PhD
Senior Editor
The EMBO Journal

Instruction for the preparation of your revised manuscript:

- 1) a .docx formatted version of the manuscript text (including legends for main figures, EV figures and tables). Please make sure that the changes are highlighted to be clearly visible.
- 2) individual production quality figure files as .eps, .tif, .jpg (one file per figure).
- 3) a .docx formatted letter INCLUDING the reviewers' reports and your detailed point-by-point response to their comments. As part of the EMBO Press transparent editorial process, the point-by-point response is part of the Review Process File (RPF), which will be published alongside your paper.
- 4) a complete author checklist, which you can download from our author guidelines ([https://wol-prod-cdn.literatumonline.com/pb-assets/embo-site/Author Checklist%20-%20EMBO%20J-1561436015657.xlsx](https://wol-prod-cdn.literatumonline.com/pb-assets/embo-site/Author%20Checklist%20-%20EMBO%20J-1561436015657.xlsx)). Please insert information in the checklist that is also reflected in the manuscript. The completed author checklist will also be part of the RPF.
- 5) Please note that all corresponding authors are required to supply an ORCID ID for their name upon submission of a revised manuscript.
- 6) It is mandatory to include a 'Data Availability' section after the Materials and Methods. Before submitting your revision, primary datasets produced in this study need to be deposited in an appropriate public database, and the accession numbers and database listed under 'Data Availability'. Please remember to provide a reviewer password if the datasets are not yet public (see <https://www.embopress.org/page/journal/14602075/authorguide#datadeposition>).
In case you have no data that requires deposition in a public database, please state so in this section. Note that the Data

Availability Section is restricted to new primary data that are part of this study.

7) Our journal encourages inclusion of *data citations in the reference list* to directly cite datasets that were re-used and obtained from public databases. Data citations in the article text are distinct from normal bibliographical citations and should directly link to the database records from which the data can be accessed. In the main text, data citations are formatted as follows: "Data ref: Smith et al, 2001" or "Data ref: NCBI Sequence Read Archive PRJNA342805, 2017". In the Reference list, data citations must be labeled with "[DATASET]". A data reference must provide the database name, accession number/identifiers and a resolvable link to the landing page from which the data can be accessed at the end of the reference. Further instructions are available at .

8) We would also encourage you to include the source data for figure panels that show essential data. Numerical data can be provided as individual .xls or .csv files (including a tab describing the data). For 'blots' or microscopy, uncropped images should be submitted (using a zip archive or a single pdf per main figure if multiple images need to be supplied for one panel). Additional information on source data and instruction on how to label the files are available at .

9) We replaced Supplementary Information with Expanded View (EV) Figures and Tables that are collapsible/expandable online (see examples in <https://www.embopress.org/doi/10.15252/emboj.201695874>). A maximum of 5 EV Figures can be typeset. EV Figures should be cited as 'Figure EV1, Figure EV2' etc. in the text and their respective legends should be included in the main text after the legends of regular figures.

11) For data quantification: please specify the name of the statistical test used to generate error bars and P values, the number (n) of independent experiments (specify technical or biological replicates) underlying each data point and the test used to calculate p-values in each figure legend. The figure legends should contain a basic description of n, P and the test applied. Graphs must include a description of the bars and the error bars (s.d., s.e.m.).

Further information is available in our Guide to Authors: <https://www.embopress.org/page/journal/14602075/authorguide>

We realize that it is difficult to revise to a specific deadline. In the interest of protecting the conceptual advance provided by the work, we recommend a revision within 3 months (16th Nov 2023). Please discuss the revision progress ahead of this time with the editor if you require more time to complete the revisions.

Referee #1:

Metabolic syndrome (MetS) is characterized by obesity, insulin resistance, hypertension and hyperlipidemia and can lead to the development of multiple diseases including type 2 diabetes, cardiovascular disease, and cancer. Despite the increasing incidence of MetS, the mechanisms contributing to the pathogenesis of MetS are still poorly understood. In the current study, Xiao and colleagues focus on investigating the role of acyl-CoA synthetase medium chain family member 3 (ACSM3) in MetS.

ACSM3 is one of a family of enzymes that catalyze the initial step in fatty acid metabolism. The authors demonstrate that loss of *Acsm3* promotes metabolic syndrome and exacerbates fatty acid, in particular lauric acid, accumulation in the liver. *Acsm3* loss was also associated with mitochondrial dysfunction in hepatocytes. Mechanistically the authors attribute mitochondrial dysfunction to lauric acid-induced activation of the transcription factor HNF4- α and subsequent induction of p38 expression. Finally, the authors provide evidence that Adezmapimod, a p38 inhibitor, can rescue mitochondrial dysfunction and MetS resulting from loss of *Acsm3*. For the most part, robust data is provided to support the conclusions. However, there are some outstanding questions that should be addressed, as outlined below.

The rationale for investigating the role of ACSM3 in MetS is based on the finding that ACSM3 levels are lower in peripheral blood of patients with MetS. ACSM3 is an outer mitochondrial membrane protein so it is not clear why ACSM3 was found in the circulation or what the significance of circulating ACSM3 is. Where do the authors propose the circulating ACSM3 is coming from? How do the authors know that lower circulating ACSM3 levels are indicative of reduced ACSM3 expression in other tissues (e.g. liver)? It is currently very difficult to know if the discoveries made using a *Acsm3* knockout mouse model would in any way be relevant to the MetS setting.

The authors show that lauric acid supplementation induces HNF4 α /p38 expression. It would be important to determine whether one of the other medium-chain fatty acids identified in Fig 2K/L or a long-chain fatty acid (e.g. palmitic acid) would have the same effect. In other words, how specific is this to phenomena to lauric acid?

Hnf4a should be included in the heatmap in Fig 4A and Hnf4a transcript expression should be examined under the conditions employed in Fig 4F/G.

Minor comments

Liver weight is higher in the *Acsm3* KO mice in Fig S2B and Fig 2C, not lower as is suggested in the text. The sentences relating to these data should be revised.

The results in the liver-specific *Acsm3* knockout model are similar to results in the systemic knockout model but they are not the same, as indicated in the text. The authors should emphasise that the impact of liver-specific *Acsm3* KO is less severe.

Hepatic acetyl-CoA content is not solely influenced by beta-oxidation so the authors should take care not to suggest that this is the case when they talk about the results shown in Fig 2J.

Zoom images, focussing on mitochondria, should be included in Fig 3B.

The Y-axis label in Fig 3C is misspelled (perimer should be perimeter)

There are numerous spelling and grammatical errors throughout the manuscript. This should be addressed before resubmission.

Journal names are missing from many of the references in the bibliography (e.g. Ref 46-54)

A previous study had identified an association between *Acsm3* expression in MetS. This study should be cited and discussed (PMID:33923085).

Referee #2:

In "ACSM3 is involved in MetS by mitochondrial dysfunction via lauric acid-HNF4 α -p38 MAPK signaling, Xiao, Li, and colleagues explore the role of the mitochondrial fatty acid metabolic enzyme ACSM3 in metabolic disease. They first observe that ACSM3 tends to be lower in peripheral blood samples from several large cohorts of patients who had metabolic syndrome, and modeled deletion of ACSM3 in mice both in whole-body and liver only, where ACSM3 is highest. In both cases, especially in mice fed a high fat and high fructose (FF) diet, there were signs of metabolic disease including elevated fatty acids and insulin resistance and impaired glucose tolerance. ACSM3 also led to general impairment of mitochondrial function. Through detailed and elegant mechanistic studies followed up in vivo, the Authors convincingly show that p38 is activated in a manner dependent on HNF4 α and stimulation by lauric acid. The Authors finally show in vivo that inhibiting p38 can rescue some of the deleterious effects of ACSM3 deletion.

Overall, this manuscript has excellent and detailed mechanistic data which are overall very convincing. The new mechanistic insights and how they connect to metabolic syndrome are significant findings. However, the paper requires moderate to extensive revision, mostly in writing and not experimentation. In many parts the data or background are not sufficiently explained. The rationale for studying ACSM3 from the human studies is unclear. The Authors also do not do a sufficient job, either in the Results or Discussion, of placing their work in the context of the wider field and citing and discussion the work of others. In particular, since it is known p38 has a role in metabolic syndrome, discussion should instead turn to whether ACSM3 itself could be drugged, or perhaps whether ACSM3 suppression is a biomarker for p38 activation. Overall, I have requested

some large-scale rewrites and a few, very small, new experiments, and look forward to reading a revised version.

Major points

- 1) It is unfortunate that most of the clinical and mouse data is done in males only, when metabolic syndrome equally affects females. This should be addressed as a major limitation of the work in the Discussion, with perhaps some discussion on whether there might be sex-specific effects in fatty acid metabolism.
- 2) I do not fully understand the logic of why the Authors chose to pursue ACSM3. They observed that it was lower in patients that already had metabolic syndrome, in peripheral blood. This seems like an effect of metabolic syndrome, since peripheral blood cells are not often directly involved in the cause of metabolic syndrome. The Authors then, inexplicably, decided to delete ACSM3 in liver, even though their initial findings had nothing to do with liver, and show that this recapitulates key features of metabolic syndrome. This makes the story very confusing with regards to causality: is ACSM3 suppression causing metabolic syndrome, or is an effect of metabolic syndrome? Why is ACSM3 lower in blood in people with metabolic syndrome? This is a problem that needs to be fixed in the Introduction and Discussion to put the findings of the paper in better context. It does not detract from the novelty of the data, but it does make the rationale difficult to follow.
- 3) It is not clear if ACSM3 deletion is causing severe liver damage, either in normal fed or FF-fed mice. Please check for levels of circulating liver markers AST and ALT in these mice in each of these conditions to quantify if liver damage is occurring.
- 4) As far as I can tell, the finding that p38 is involved in metabolic syndrome and insulin resistance is not novel and has been described by many others. This is OK, the novelty here is that p38 is upregulated downstream of ACSM3 deletion, but the Authors must discuss other prior literature on p38 and its connection to metabolic syndrome, both in the Results and the Discussion section.
- 5) The Discussion section is not acceptable. It is short, and mostly revisits the major findings of the paper, with very little space devoted to how their work fits in the with the field. The Discussion section should be expanded to discuss more literature on these topics from other labs.

Minor points

- 1) Does the "second cohort for validation" have males or females, or just males? This is unclear and needs to be in the paper.
- 2) Also about the second cohort: the description in the Methods and Results are different. In the Methods, they are described as 32-84 years, but in the Results (page 15), they are described as 60-75 years. Is this just a subset of the second cohort? Why was this age range chosen? This needs to be in the paper.
- 3) Please define HOMA-IR in the Results section the first time it is mentioned.
- 4) Since a large amount of space in the Results section is devoted to Supplemental Figure S2 and the data are very important, it should be a main figure.
- 5) Please quantify all IHC images (Figure 2G-H, Figure 5 F-G, etc), and perform statistical analysis on multiple fields.
- 6) On Page 17, please devote one paragraph to describing the results of Supplemental Figure S4 and S5, or remove them from the paper if they do not need to be there.
- 7) Please refer to adezmapimod the same way in the Results and in the Figures (either adezmapimod in both sections, or SB203580 in both sections).

Referee #1:

Metabolic syndrome (MetS) is characterized by obesity, insulin resistance, hypertension and hyperlipidemia and can lead to the development of multiple diseases including type 2 diabetes, cardiovascular disease, and cancer. Despite the increasing incidence of MetS, the mechanisms contributing to the pathogenesis of MetS are still poorly understood. In the current study, Xiao and colleagues focus on investigating the role of acyl-CoA synthetase medium-chain family member 3 (ACSM3) in MetS. ACSM3 is one of a family of enzymes that catalyze the initial step in fatty acid metabolism. The authors demonstrate that loss of Acsm3 promotes metabolic syndrome and exacerbates fatty acid, in particular lauric acid, accumulation in the liver. Acsm3 loss was also associated with mitochondrial dysfunction in hepatocytes. Mechanistically the authors attribute mitochondrial dysfunction to lauric acid-induced activation of the transcription factor HNF4-alpha and subsequent induction of p38 expression. Finally, the authors provide evidence that Adezmapimod, a p38 inhibitor, can rescue mitochondrial dysfunction and MetS resulting from loss of Acsm3. For the most part, robust data is provided to support the conclusions. However, there are some outstanding questions that should be addressed, as outlined below.

The rationale for investigating the role of ACSM3 in MetS is based on the finding that ACSM3 levels are lower in peripheral blood of patients with MetS. ACSM3 is an outer mitochondrial membrane protein so it is not clear why ACSM3 was found in the circulation or what the significance of circulating ACSM3 is. Where do the authors propose the circulating ACSM3 is coming from? How do the authors know that lower circulating ACSM3 levels are indicative of reduced ACSM3 expression in other tissues (e.g. liver)? It is currently very difficult to know if the discoveries made using a Acsm3 knockout mouse model would in any way be relevant to the MetS setting.

Our response: I apologize for the confusion. The description of the rationale was revised to make it clear, which is shown as below.

To gain deep insight into the pathogenesis of metabolic syndrome (MetS), we first recruited an essential MetS cohort with strict criteria and performed genome-wide transcriptome analysis in peripheral blood. ACSM3 was found to be significantly lower expressed in the MetS patients, which was validated in another larger cohort (**Fig 1A-C**). Lower expression of Acsm3 was also observed in the peripheral blood of MetS mice (**Fig 1D**). Thus, we confirmed that ACSM3 was associated with MetS, but the underlying mechanism needed to be elucidated. First, we analyzed the tissue distribution of Acsm3 in mice, and high expression was found in the liver, which is an important organ in glucose and lipid metabolism (**Fig 1E and F**). Second, we focused on the liver and found that Acsm3 was also expressed at lower levels in the livers of MetS mice (**Fig 1G and H**). Thus, we then used an Acsm3 knockout mouse model to explore the participation of Acsm3 in MetS.

Circulating ACSM3 mainly comes from leukocytes. The lower circulating ACSM3 levels give us some clues, and we further confirmed that Acsm3 (highly expressed in

the liver) was also expressed at lower levels in the livers of MetS mice by RT-qPCR and Western blot assays (**Fig 1G and H**). This is a good question about the significance of circulating ACSM3, which will be the focus of our future studies.

Fig 1 ACSM3 expression in metabolic syndrome patients and mice. (A) Volcano plot showing the gene expression from whole blood cells between 18 healthy subjects and 51 metabolic syndrome (MetS) patients. *x*-axis: \log_2FC , Fold change (unpaired *t* test) displayed on a \log_2 scale. *y*-axis: $-\log_{10}(P \text{ value})$. (B) The relative mRNA expression of *ACSM3* verified in whole blood cells in the above cohort using a real-time quantitative PCR (RT-qPCR) assay. (C) The relative mRNA expression of *ACSM3* verified in the second cohort. A total of 386 subjects were classified with MetS, and 440 were classified as control subjects. (D) The relative mRNA expression of *Acsm3* in the peripheral blood of control and MetS mice ($n = 6$ biologically independent samples in each group). (E) The relative mRNA expression of *Acsm3* in different tissues of mice ($n = 6$ biologically independent samples in each group). (F) Western blot and quantification of the relative protein expression of *Acsm3* in different tissues of mice. Three results were used for quantification, $n = 3$ biologically independent samples in each group. (G) The relative mRNA expression of *Acsm3* in the livers of control and MetS mice ($n = 6$ biologically independent samples in each group). (H) Western blot and quantification of the relative protein expression of *Acsm3* in the livers of control and MetS mice. Three results were used for quantification, $n = 6$ biologically independent samples in each group. Values are represented as the mean \pm SD. Statistics were performed using Student's *t* test. The *P* values are shown in the figure.

The authors show that lauric acid supplementation induces HNF4 α /p38 expression. It would be important to determine whether one of the other medium-chain fatty acids identified in Fig 2K/L or a long-chain fatty acid (e.g. palmitic acid) would have the same effect. In other words, how specific is this to phenomena to lauric acid?

Our response: Thank you for the comment. After *Acsm3* knockout, lauric acid accumulated the most among all detectable medium-chain fatty acids. Thus, lauric acid was specific for *Acsm3*, and we focused on it.

It would be important to determine whether one of the other medium-chain fatty acids or a long-chain fatty acid would have the same effect, however, that is less relevant for understanding *Acsm3* biology and is not the focus of this study. Thank you for this good question, and we will explore it in the future.

Hnf4a should be included in the heatmap in Fig 4A and *Hnf4a* transcript expression should be examined under the conditions employed in Fig 4F/G.

Our response: Thank you for your suggestions. The newly added results are shown in Fig 5A, F, and G.

Fig 5A Heatmap showing the relative expression of hepatic DEGs (unpaired t test, *Acsm3* KO vs control mice, fed an FF diet for eight weeks) enriched in the p38MAPK

cascade pathway and *Hnf4a*, $n = 3$ biologically independent samples in each group.

A

Fig 5F and G The relative mRNA expression of *Hnf4a* was detected using RT-qPCR assay. $n = 6$ biologically independent samples in each group.

F

G

Minor comments

Liver weight is higher in the Acsm3 KO mice in Fig S2B and Fig 2C, not lower as is suggested in the text. The sentences relating to these data should be revised.

Our response: Sorry for the confusion. The revised sentences are as follows:

“...the body weight was lower, while the liver weight and liver/body weight ratio were higher when compared with controls (Fig 2D, previous Fig S2B).”

“When fed FF, *Acsm3* knockout mice also presented lower body weight, higher liver weight, and a higher liver/body weight ratio (Fig 3A, previous Fig 2C).”

The results in the liver-specific Acsm3 knockout model are similar to results in the systemic knockout model but they are not the same, as indicated in the text. The authors should emphasize that the impact of liver-specific Acsm3 KO is less severe.

Our response: Sorry for the confusion. Under the same diet, systemic or liver-specific *Acsm3* knockout mice had similar effects on metabolic syndrome. These results emphasized the functions of hepatic *Acsm3*.

Fig 2 shows the effect of **systemic *Acsm3* knockout** under a **normal diet**.

Fig EV3 shows the effect of **liver-specific *Acsm3* knockout** under a **normal diet**.

Fig 3 shows the effect of **systemic *Acsm3* knockout** under an **FF diet**.

Fig EV4 shows the effect of **liver-specific *Acsm3* knockout** under an **FF diet**.

Hepatic acetyl-CoA content is not solely influenced by beta-oxidation so the authors should take care not to suggest that this is the case when they talk about the results shown in Fig 2J.

Our response: Thank you for the suggestion. We deleted the related description of beta-oxidation. The revised sentence is as follows:

“Furthermore, we found that the acetyl-CoA level was significantly decreased in the knockout mice fed an FF diet (Fig 3I).”

Zoom images, focusing on mitochondria, should be included in Fig 3B.

Our response: Thank you for your suggestion. We replaced them with clearer and more representative images. The zoom images are shown in **Fig 4B**.

Fig 4B Transmission electron microscopy (TEM) photographs of liver tissues from *Acsm3* KO and control mice. Scale bar, 1 μ m.

The Y-axis label in Fig 3C is misspelled (perimer should be perimeter)

Our response: We apologize for the typos. The revised panel is shown in **Fig 4C**.

Fig 4C Distribution of the mitochondrial perimeter (μ m) in the livers of *Acsm3* KO and control mice.

There are numerous spelling and grammatical errors throughout the manuscript. This should be addressed before resubmission.

Our response: Sorry for the typos and grammatical errors, and one native speaker has

helped us to revise the whole manuscript.

Journal names are missing from many of the references in the bibliography (e.g. Ref 46-54)

Our response: Sorry for the mistakes. We double-checked and corrected all the reference formats.

A previous study had identified an association between Acsm3 expression in MetS. This study should be cited and discussed (PMID:33923085)

Our response: Thank you for your suggestion. We cited and discussed this study (PMID:33923085) in our revised manuscript as follows:

“In a recent study, Junková *et al.* compared the hepatic transcriptome of spontaneously hypertensive rat, polydactylous rat (PD, an animal model of hypertriglyceridemia, IR, and obesity), and Brown Norway strain that is resistant to MetS development (Junková *et al.*, 2021). As they said, hepatic *Acsm3* was also markedly downregulated in PD strains, which is consistent with our findings.”

We greatly appreciate your helpful and valuable comments, and we would like to thank you for your valuable time and effort in reviewing our revised manuscript.

Referee #2:

In "ACSM3 is involved in MetS by mitochondrial dysfunction via lauric acid-HNF4a-p38 MAPK signaling, Xiao, Li, and colleagues explore the role of the mitochondrial fatty acid metabolic enzyme ACSM3 in metabolic disease. They first observe that ACSM3 tends to be lower in peripheral blood samples from several large cohorts of patients who had metabolic syndrome, and modeled deletion of ACSM3 in mice both in whole-body and liver only, where ACSM3 is highest. In both cases, especially in mice fed a high fat and high fructose (FF) diet, there were signs of metabolic disease including elevated fatty acids and insulin resistance and impaired glucose tolerance. ACSM3 also led to general impairment of mitochondrial function. Through detailed and elegant mechanistic studies followed up in vivo, the Authors convincingly show that p38 is activated in a manner dependent on HNF4a and stimulation by lauric acid. The Authors finally show in vivo that inhibiting p38 can rescue some of the deleterious effects of ACSM3 deletion.

Overall, this manuscript has excellent and detailed mechanistic data which are overall very convincing. The new mechanistic insights and how they connect to metabolic syndrome are significant findings. However, the paper requires moderate to extensive revision, mostly in writing and not experimentation. In many parts the data or background are not sufficiently explained. The rationale for studying ACSM3 from the human studies is unclear. The Authors also do not do a sufficient job, either in the Results or Discussion, of placing their work in the context of the wider field and citing and discussion the work of others. In particular, since it is known p38 has a role in metabolic syndrome, discussion should instead turn to whether ACSM3 itself could be drugged, or perhaps whether ACSM3 suppression is a biomarker for p38 activation. Overall, I have requested some large-scale rewrites and a few, very small, new experiments, and look forward to reading a revised version.

Major points

1) It is unfortunate that most of the clinical and mouse data is done in males only, when metabolic syndrome equally affects females. This should be addressed as a major limitation of the work in the Discussion, with perhaps some discussion on whether there might be sex-specific effects in fatty acid metabolism.

Our response: Thank you for the suggestion. When we performed the animal experiments, we took into account that male mice are more sensitive to diet-induced MetS (metabolic syndrome) than female mice according to previous studies (Kleinert *et al*, 2018; Leonardi *et al*, 2020), and thus, male mice were used in this study. However, sex differences may lead to experimental bias, and we discussed this limitation as follows:

“The major limitation is that mouse data were only collected in males when MetS equally affects females. Some evidence suggests sex differences in FA metabolism (Adler-Wailes *et al*, 2013; Blaak, 2001; Decsi & Kennedy, 2011; Jensen, 1995). Adult women release more free FA relative to resting energy expenditure and

have greater free FA clearance rates than men(Adler-Wailes *et al*, 2013). In adolescents, independent of the effects of resting energy expenditure and fat mass, free FA kinetics differ significantly in obese adolescent girls and boys, with greater free FA flux among girls(Adler-Wailes *et al*, 2013). Furthermore, sex hormones may influence the enzymatic synthesis of long-chain polyunsaturated FAs, which may lead to sex-specific differences in long-chain polyunsaturated FA status(Decsi & Kennedy, 2011). A review of the literature generally suggested that there was a higher contribution of arachidonic acid and docosahexaenoic acid in blood lipids in women than in men; however, sex-specific differences were not seen in every study(Blaak, 2001; Decsi & Kennedy, 2011; Jensen, 1995). Further experimental studies are needed to elucidate the effects of sex differences on Acsm3 participation in FA metabolism.”

2) I do not fully understand the logic of why the Authors chose to pursue ACSM3. They observed that it was lower in patients that already had metabolic syndrome, in peripheral blood. This seems like an effect of metabolic syndrome, since peripheral blood cells are not often directly involved in the cause of metabolic syndrome. The Authors then, inexplicably, decided to delete ACSM3 in liver, even though their initial findings had nothing to do with liver, and show that this recapitulates key features of metabolic syndrome. This makes the story very confusing with regards to causality: is ACSM3 suppression causing metabolic syndrome, or is an effect of metabolic syndrome? Why is ACSM3 lower in blood in people with metabolic syndrome? This is a problem that needs to be fixed in the Introduction and Discussion to put the findings of the paper in better context. It does not detract from the novelty of the data, but it does make the rationale difficult to follow.

Our response: Sorry for the confusion. We revised the related description, which makes the logic much clearer.

To gain deep insight into the pathogenesis of metabolic syndrome (MetS), we first recruited an essential MetS cohort with strict criteria and performed genome-wide transcriptome analysis in peripheral blood. *ACSM3* was found to be significantly lower expressed in the MetS patients, which was validated in another larger cohort (**Fig 1A-C**). Lower expression of *Acsm3* was also observed in the peripheral blood of MetS mice (**Fig 1D**). Thus, we confirmed that *ACSM3* was associated with MetS, but the underlying mechanism needed to be elucidated. First, we analyzed the tissue distribution of *Acsm3* in mice, and high expression was found in the liver, which is an important organ in glucose and lipid metabolism (**Fig 1E and F**). Second, we focused on the liver and found that *Acsm3* was also expressed at lower levels in the livers of MetS mice (**Fig 1G and H**). Thus, we then used an *Acsm3* knockout mouse model to explore the participation of *Acsm3* in MetS.

Fig 1 ACSM3 expression in metabolic syndrome patients and mice. (A) Volcano plot showing the gene expression from whole blood cells between 18 healthy subjects and 51 metabolic syndrome (MetS) patients. *x*-axis: \log_2FC , Fold change (unpaired *t* test) displayed on a \log_2 scale. *y*-axis: $-\log_{10}(P \text{ value})$. (B) The relative mRNA

expression of *ACSM3* verified in whole blood cells in the above cohort using a real-time quantitative PCR (RT-qPCR) assay. (C) The relative mRNA expression of *ACSM3* verified in the second cohort. A total of 386 subjects were classified with MetS, and 440 were classified as control subjects. (D) The relative mRNA expression of *Acsm3* in the peripheral blood of control and MetS mice ($n = 6$ biologically independent samples in each group). (E) The relative mRNA expression of *Acsm3* in different tissues of mice ($n = 6$ biologically independent samples in each group). (F) Western blot and quantification of the relative protein expression of *Acsm3* in different tissues of mice. Three results were used for quantification, $n = 3$ biologically independent samples in each group. (G) The relative mRNA expression of *Acsm3* in the livers of control and MetS mice ($n = 6$ biologically independent samples in each group). (H) Western blot and quantification of the relative protein expression of *Acsm3* in the livers of control and MetS mice. Three results were used for quantification, $n = 6$ biologically independent samples in each group. Values are represented as the mean \pm SD. Statistics were performed using Student's t test. The P values are shown in the figure.

3) It is not clear if *ACSM3* deletion is causing severe liver damage, either in normal fed or FF-fed mice. Please check for levels of circulating liver markers AST and

ALT in these mice in each of these conditions to quantify if liver damage is occurring.

Our response: Thank you for your comments. Under a normal diet, we found that there were no significant differences in the circulating ALT and AST levels between *Acsm3* knockout (or liver-specific knockout) and control mice (**Fig 2K** and **Fig EV3H**). However, under the FF diet, *Acsm3* deletion (or liver-specific deletion) significantly upregulated circulating ALT and AST levels (**Fig 3E** and **Fig EV4H**).

Fig 2K The contents (IU/mL) of serum ALT and AST in *Acsm3* KO and control mice fed a normal diet ($n = 6$ biologically independent samples in each group).

Fig EV3H The contents (IU/mL) of serum ALT and AST in sh*Acsm3* and shControl mice fed a normal diet ($n = 6$ biologically independent samples in each group).

Fig 3E The contents (IU/mL) of serum ALT and AST in *Acsm3* KO and control mice fed the FF diet ($n = 6$ biologically independent samples in each group).

Fig EV4H The contents (IU/mL) of serum ALT and AST in sh*Acsm3* and shControl mice fed the FF diet ($n = 6$ biologically independent samples in each group).

H

4) As far as I can tell, the finding that p38 is involved in metabolic syndrome and insulin resistance is not novel and has been described by many others. This is OK, the novelty here is that p38 is upregulated downstream of ACSM3 deletion, but the Authors must discuss other prior literature on p38 and its connection to metabolic syndrome, both in the Results and the Discussion section.

Our response: Thank you for your comments. We discussed this in the Results and Discussion sections as follows:

In Results:

“P38 mitogen-activated protein kinase (p38 MAPK) signaling is closely related to a variety of intracellular responses, including inflammation, oxidative stress, ROS, and apoptosis, which can participate in MetS by multiple approaches(Chan *et al*, 2019; Woo *et al*, 2023). Noticeably, previous studies have shown that p38 MAPK (MAPK14) is a crucial player in mitochondrial dysfunction(Huang *et al*, 2018; Huang *et al*, 2013; Manikanta *et al*, 2020). Excessive activation of the p38 MAPK pathway results in disrupting mitochondrial membrane potential and releasing ROS(Huang *et al*, 2018). Inhibition of the p38 MAPK pathway could attenuate mitochondrial dysfunction(Huang *et al*, 2018; Kamoshita *et al*, 2022; Kumphune *et al*, 2015). Therefore, we hypothesized that p38 MAPK may be a vital factor involved in hepatic mitochondrial dysfunction and MetS in Acsm3 knockout mice.”

In Discussion:

“Sufficient evidence indicates an association between p38 MAPK and MetS(Fang *et al*, 2019; Lan *et al*, 2022; Wang *et al*, 2020; Wu *et al*, 2021; Zhang *et al*, 2019). P38 MAPK regulates inflammatory activation in metabolic hepatic disease. Macrophage p38 α promotes the progression of steatohepatitis by inducing pro-inflammatory cytokine (such as MIP-2 α , IL-1 β , CXCL10, and IL-6) secretion and M1 polarization(Zhang *et al*, 2019). P38 MAPK is also involved in arginase-II-mediated endothelial nitric oxide synthase uncoupling in a high-fat diet-induced obesity mouse model, which links obesity-associated IR and type-II diabetes to the increased incidence of cardiovascular disease(Yu *et al*, 2014). Furthermore, elevated phosphorylation of p38 MAPK is observed in the livers of diabetic mice(Qiao *et al*, 2006). Its synergistic effects with the phosphorylation of cAMP-response element binding protein enhance hepatic PGC-1 α expression to increase hepatic gluconeogenesis(Fang *et al*, 2019). P38 MAPK activation can also result in mitochondrial dysfunction through increasing ROS levels and oxidative stress(Liu *et*

al, 2022; Manikanta *et al*, 2020; Wang *et al*, 1992). In our study, the activation of p38 MAPK signal transduction explains the abnormal phenotype of glucose/lipid metabolism and mitochondrial dysfunction caused by *Acsm3* knockout.”

5) *The Discussion section is not acceptable. It is short, and mostly revisits the major findings of the paper, with very little space devoted to how their work fits in the with the field. The Discussion section should be expanded to discuss more literature on these topics from other labs.*

Our response: Thank you for your comments. We rewrote the discussion. The devotion of our work in the field, the limitations, and more literature on these topics were discussed in the revised version as follows:

Discussion

In this study, we identified ACSM3, a vital member of the ACSM family, whose deficiency promotes MetS progression. We determined that lauric acid was the specific FA catalyzed by *Acsm3*, and the mitochondrial dysfunction and metabolic disorders caused by *Acsm3* were attributed to its induced activation of the lauric acid/Hnf4 α /p38 MAPK pathway. Our study provided novel targets for mechanism decryption and clinical therapy of MetS (Fig 6O).

Acyl-CoA synthetases (ACSSs) are essential for *de novo* lipid synthesis, FA catabolism, and membrane remodeling(Soupene & Kuypers, 2008). ACSs perform the initial reaction for cellular FA metabolism by ligating a coenzyme A to an FA, which both traps an FA within a cell and activates it for metabolism(Fernandez & Ellis, 2020). The ACS family of enzymes is large and can be broken into subfamilies termed short-, medium-, long-, and very-long-chain ACSs, each with unique distribution across and within cell types and differential FA substrate preferences(Fernandez & Ellis, 2020; Watkins *et al*, 2007). ACSM3 is a member of the ACSM family, which was identified to be lower expressed in MetS patients and mice in our study. In a recent study, Junková *et al*. compared the hepatic transcriptome of spontaneously hypertensive rat, polydactylous rat (PD, an animal model of hypertriglyceridemia, IR, and obesity), and Brown Norway strain that is resistant to MetS development(Junková *et al*, 2021). As they said, hepatic *Acsm3* was also markedly downregulated in PD strains, which is consistent with our findings.

Medium-chain FAs have been used as an energy source of enteric nutrients because of their rapid metabolism(Schönfeld & Wojtczak, 2016). Medium-chain FAs are directly transported to the liver after passing through the portal vein and are taken up by mitochondria in a carnitine-independent pathway(Kamoshita *et al*, 2022; Nagao & Yanagita, 2010). Overload of medium-chain FAs, particularly in the liver, promotes FA synthesis, induces hepatic steatosis, and causes IR(Turner *et al*, 2009). We found that hepatic medium-chain FAs significantly accumulated in *Acsm3* knockout mice. Excess FAs were esterified to more glycerol3-phosphate and cholesterol to produce TG or cholesteryl esters, respectively. These neutral lipids can be either stored in cytoplasmic lipid droplets or secreted into the bloodstream as very low-density lipoprotein particles(Alves-Bezerra & Cohen, 2017). Notably, free FAs could also

induce IR through intrahepatocellular activation of several serine/threonine kinases, reduction in tyrosine phosphorylation of the insulin receptor substrate (IRS)-1/2, and impairment of the IRS/PI3K pathway of insulin signaling(Chai *et al*, 2011).

Lauric acid has been reported to participate in multiple metabolic and cardiovascular diseases(Kamoshita *et al*, 2022; Miyake *et al*, 2022; Saraswathi *et al*, 2020). It should be noted that coconut oil, which contains 50% lauric acid as the primary fatty acid, induces IR in the liver but not in skeletal muscle and adipose tissue(Turner *et al*, 2009). Lauric acid could also upregulate *Hnf4 α* , which activated Selenop transcription and repaired insulin-induced Akt phosphorylation(Kamoshita *et al*, 2022). Furthermore, heart failure patients with MetS displayed a higher proportion of lauric acid than heart failure patients without MetS(Lee *et al*, 2012). In a nonalcoholic fatty liver disease (NAFLD) cohort, higher lauric acid was significantly associated with a high NAFLD activity score in analyses adjusted for the same factors and fibrosis stage(Miyake *et al*, 2022). In our study, lauric acid was highly accumulated in the livers of *Acsm3* knockout mice, playing a crucial role in causing the MetS phenotype.

Sufficient evidence indicates an association between p38 MAPK and MetS(Fang *et al*, 2019; Lan *et al*, 2022; Wang *et al*, 2020; Wu *et al*, 2021; Zhang *et al*, 2019). P38 MAPK regulates inflammatory activation in metabolic hepatic disease. Macrophage p38 α promotes the progression of steatohepatitis by inducing pro-inflammatory cytokine (such as MIP-2 α , IL-1 β , CXCL10, and IL-6) secretion and M1 polarization(Zhang *et al*, 2019). P38 MAPK is also involved in arginase-II-mediated endothelial nitric oxide synthase uncoupling in a high-fat diet-induced obesity mouse model, which links obesity-associated IR and type-II diabetes to the increased incidence of cardiovascular disease(Yu *et al*, 2014). Furthermore, elevated phosphorylation of p38 MAPK is observed in the livers of diabetic mice(Qiao *et al*, 2006). Its synergistic effects with the phosphorylation of cAMP-response element binding protein enhance hepatic PGC-1 α expression to increase hepatic gluconeogenesis(Fang *et al*, 2019). P38 MAPK activation can also result in mitochondrial dysfunction through increasing ROS levels and oxidative stress(Liu *et al*, 2022; Manikanta *et al*, 2020; Wang *et al*, 1992). In our study, the activation of p38 MAPK signal transduction explains the abnormal phenotype of glucose/lipid metabolism and mitochondrial dysfunction caused by *Acsm3* knockout.

There are some limitations in this study. The major limitation is that mouse data were only collected in males when MetS equally affects females. Some evidence suggests sex differences in FA metabolism(Adler-Wailes *et al*, 2013; Blaak, 2001; Decsi & Kennedy, 2011; Jensen, 1995). Adult women release more free FA relative to resting energy expenditure and have greater free FA clearance rates than men(Adler-Wailes *et al*, 2013). In adolescents, independent of the effects of resting energy expenditure and fat mass, free FA kinetics differ significantly in obese adolescent girls and boys, with greater free FA flux among girls(Adler-Wailes *et al*, 2013). Furthermore, sex hormones may influence the enzymatic synthesis of long-chain polyunsaturated FAs, which may lead to sex-specific differences in long-chain polyunsaturated FA status(Decsi & Kennedy, 2011). A review of the

literature generally suggested that there was a higher contribution of arachidonic acid and docosahexaenoic acid in blood lipids in women than in men; however, sex-specific differences were not seen in every study (Blaak, 2001; Decsi & Kennedy, 2011; Jensen, 1995). Further experimental studies are needed to elucidate the effects of sex differences on *Acs3* participation in FA metabolism. Another limitation is that although lauric acid is the most specific medium-chain FA catalyzed by *Acs3*, multiple other FAs also accumulated after *Acs3* knockout. We cannot determine whether one of the other FAs would have the same or even more severe effect as lauric acid (such as activating p38 MAPK), which should be explored in further studies.

In summary, we clarified the mechanism by which *Acs3* is involved in MetS. *Acs3* deletion caused lauric acid accumulation, further promoting *Hnf4a* and *Mapk14* transcriptions to activate p38 MAPK signaling, which impaired mitochondrial function and contributed to abnormal glucose and lipid metabolism. The p38 MAPK inhibitor adempagimod could rescue the metabolic disorders caused by *Acs3* deficiency. Taken together, these findings provided novel insights into the roles of ACSM3 in MetS and hinted at new therapeutic strategies.

Minor points

1) Does the "second cohort for validation" have males or females, or just males? This is unclear and needs to be in the paper.

Our response: Sorry for the confusion. In the second cohort, 48.2% (212/440) of control individuals were male. A total of 51.8% (200/386) of MetS patients were male. We provided this information in the Result section.

2) Also about the second cohort: the description in the Methods and Results are different. In the Methods, they are described as 32-84 years, but in the Results (page 15), they are described as 60-75 years. Is this just a subset of the second cohort? Why was this age range chosen? This needs to be in the paper.

Our response: We apologize for the typos. The individuals in the second cohort were aged 60-75 years. We corrected this in the Methods.

3) Please define HOMA-IR in the Results section the first time it is mentioned.

Our response: Thank you for your suggestion. We defined HOMA-IR = fasting Glu (mmol/L) × fasting Ins (μU/mL)/22.5 in the Results section for the first time as follows:

“To measure the effects of *Acs3* on glucose homeostasis, insulin tolerance tests (ITT), glucose tolerance tests (GTT), glucose and insulin measurement, and homeostasis model assessment of insulin resistance (HOMA-IR, = fasting Glu (mmol/L) × fasting Ins (μU/mL)/22.5) calculations were conducted.”

4) Since a large amount of space in the Results section is devoted to Supplemental Figure S2 and the data are very important, it should be a main figure.

Our response: Figure S2 is placed in our revised manuscript as Figure 2.

5) Please quantify all IHC images (Figure 2G-H, Figure 5 F-G, etc.), and perform statistical analysis on multiple fields.

Our response: The qualifications and statistical analysis are shown in Fig 3F, G and Fig 6F, G.

Fig 3F, G Representative images of H&E and oil red O staining of *Acsm3* KO and control mouse livers. Scale bar, 50 μ m. The quantification of the relative area of ballooning degeneration or lipid droplets was based on NIH ImageJ software. Values are represented as the mean \pm SD ($n = 6$ biologically independent samples in each group). Statistics were performed using Student's t test.

Fig 6F, G Representative images of H&E and oil red O staining. Scale bar, 50 μ m. The quantification of the relative area of ballooning degeneration or lipid droplets was based on NIH ImageJ software. Values are represented as the mean \pm SD ($n = 6$ biologically independent samples in each group). Statistics were performed using Student's t test.

6) On Page 17, please devote one paragraph to describing the results of Supplemental Figure S4 and S5, or remove them from the paper if they do not need to be there.

Our response: We described the results of Figures S4 and S5 (called Fig EV3 and Fig EV4 in the revised version) as follows:

“The MetS phenotype was detected in whole-body *Acsm3* knockout mice, but to be more rigorous, we created a liver-specific virus to exclusively knock down *Acsm3* in the liver and found similar results (Fig EV3A-J and Fig EV4A-J). Liver-specific *Acsm3* knockdown mice revealed mild metabolic abnormalities with ND, consistent with the phenotypes caused by the systematic *Acsm3* knockout mice (Fig 2A-M). After hepatic *Acsm3* deficiency, the mice exhibited impaired glucose tolerance and insulin tolerance and increased HOMA-IR indexes (Fig EV3D-G). In addition, their hepatic and serum TG contents were also upregulated (Fig EV3I and J). After 12 weeks of FF feeding, the hepatic knockdown mice showed a more severe MetS

phenotype similar to systematic Acsm3 deletion, which illustrated the crucial role of hepatic Acsm3 in promoting MetS (Figure 3A-H and Fig EV4C-J).”

7) Please refer to adezmapimod the same way in the Results and in the Figures (either adezmapimod in both sections, or SB203580 in both sections).

Our response: We consistently refer to “adezmapimod” in our revised manuscript.

We greatly appreciate your helpful and valuable comments, and we would like to thank you for your valuable time and effort in reviewing our revised manuscript.

References

- Adler-Wailes DC, Periwai V, Ali AH, Brady SM, McDuffie JR, Uwaifo GI, Tanofsky-Kraff M, Salaita CG, Hubbard VS, Reynolds JC, Chow CC, Sumner AE, Yanovski JA (2013) Sex-associated differences in free fatty acid flux of obese adolescents. *J Clin Endocrinol Metab* 98: 1676-84
- Alves-Bezerra M, Cohen DE (2017) Triglyceride Metabolism in the Liver. In *Comprehensive Physiology*, pp 1-22.
- Blaak E (2001) Gender differences in fat metabolism. *Current Opinion in Clinical Nutrition and Metabolic Care* 4: 499-502
- Chai W, Liu J, Jahn LA, Fowler DE, Barrett EJ, Liu Z (2011) Salsalate attenuates free fatty acid-induced microvascular and metabolic insulin resistance in humans. *Diabetes Care* 34: 1634-8
- Chan CM, Huang DY, Sekar P, Hsu SH, Lin WW (2019) Reactive oxygen species-dependent mitochondrial dynamics and autophagy confer protective effects in retinal pigment epithelial cells against sodium iodate-induced cell death. *J Biomed Sci* 26: 40
- Decsi T, Kennedy K (2011) Sex-specific differences in essential fatty acid metabolism. *The American Journal of Clinical Nutrition* 94: S1914-S1919
- Fang P, Sun Y, Gu X, Shi M, Bo P, Zhang Z, Bu L (2019) Baicalin ameliorates hepatic insulin resistance and gluconeogenic activity through inhibition of p38 MAPK/PGC-1 α pathway. *Phytomedicine* 64: 153074
- Fernandez RF, Ellis JM (2020) Acyl-CoA synthetases as regulators of brain phospholipid acyl-chain diversity. *Prostaglandins, Leukotrienes and Essential Fatty Acids* 161: 102175
- Huang L, Hou Y, Wang L, Xu X, Guan Q, Li X, Chen Y, Zhou W (2018) p38 Inhibitor Protects Mitochondrial Dysfunction by Induction of DJ-1 Mitochondrial Translocation After Subarachnoid Hemorrhage. *Journal of Molecular Neuroscience* 66: 163-171
- Huang L, Wan J, Chen Y, Wang Z, Hui L, Li Y, Xu D, Zhou W (2013) Inhibitory effects of p38 inhibitor against mitochondrial dysfunction in the early brain injury after subarachnoid hemorrhage in mice. *Brain Research* 1517: 133-140
- Jensen MD (1995) Gender differences in regional fatty acid metabolism before and after meal ingestion. *The Journal of Clinical Investigation* 96: 2297-2303
- Junková K, Mirchi LF, Chylíková B, Janků M, Šilhavý J, Hüttl M, Marková I, Miklánková D, Včelák J, Malínská H, Pravenec M, Šeda O, Liška F (2021) Hepatic Transcriptome Profiling Reveals Lack of Acs3 Expression in Polydactylous Rats with High-Fat Diet-Induced Hypertriglyceridemia and Visceral Fat Accumulation. *Nutrients* 13: 1462
- Kamoshita K, Tsugane H, Ishii K-A, Takayama H, Yao X, Abuduwaili H, Tanida R, Taniguchi Y, Oo HK, Gafiyatullina G, Kaneko S, Matsugo S, Takamura T (2022) Lauric acid impairs insulin-induced Akt phosphorylation by upregulating SELENOP expression via HNF4 α induction. *American Journal of Physiology-Endocrinology and Metabolism* 322: E556-E568
- Kleinert M, Clemmensen C, Hofmann SM, Moore MC, Renner S, Woods SC, Huypens P, Beckers J, de Angelis MH, Schürmann A, Bakhti M, Klingenspor M, Heiman M, Cherrington AD, Ristow M, Lickert H, Wolf E, Havel PJ, Müller TD, Tschöp MH (2018) Animal models of obesity and diabetes mellitus. *Nature Reviews Endocrinology* 14: 140-162
- Kumphune S, Surinkaew S, Chattipakorn SC, Chattipakorn N (2015) Inhibition of p38 MAPK activation protects cardiac mitochondria from ischemia/reperfusion injury. *Pharmaceutical Biology* 53: 1831-1841
- Lan T, Hu Y, Hu F, Li H, Chen Y, Zhang J, Yu Y, Jiang S, Weng Q, Tian S, Ma T, Yang G, Luo D, Wang

- L, Li K, Piao S, Rong X, Guo J (2022) Hepatocyte glutathione S-transferase mu 2 prevents non-alcoholic steatohepatitis by suppressing ASK1 signaling. *Journal of Hepatology* 76: 407-419
- Lee S, Do HJ, Kang S-M, Chung JH, Park E, Shin M-J (2012) Plasma phospholipid fatty acid composition and estimated desaturase activity in heart failure patients with metabolic syndrome. *Journal of Clinical Biochemistry and Nutrition* 51: 150-155
- Leonardi BF, Gosmann G, Zimmer AR (2020) Modeling Diet-Induced Metabolic Syndrome in Rodents. *Molecular Nutrition & Food Research* 64: e2000249
- Liu J, Wei Y, Jia W, Can C, Wang R, Yang X, Gu C, Liu F, Ji C, Ma D (2022) Chenodeoxycholic acid suppresses AML progression through promoting lipid peroxidation via ROS/p38 MAPK/DGAT1 pathway and inhibiting M2 macrophage polarization. *Redox Biology* 56: 102452
- Manikanta K, Naveen Kumar SK, Hemshekhar M, Kemparaju K, Girish KS (2020) ASK1 inhibition triggers platelet apoptosis via p38-MAPK-mediated mitochondrial dysfunction. *Haematologica* 105: e419-e423
- Miyake T, Furukawa S, Matsuura B, Yoshida O, Miyazaki M, Shiomi A, Kanzaki S, Nakaguchi H, Sunago K, Nakamura Y, Imai Y, Watanabe T, Yamamoto Y, Koizumi Y, Tokumoto Y, Hirooka M, Kumagi T, Abe M, Hiasa Y (2022) Plasma Fatty Acid Composition Is Associated with Histological Findings of Nonalcoholic Steatohepatitis. *Biomedicines* 10: 2540
- Nagao K, Yanagita T (2010) Medium-chain fatty acids: Functional lipids for the prevention and treatment of the metabolic syndrome. *Pharmacological Research* 61: 208-212
- Qiao L, MacDougald OA, Shao J (2006) CCAAT/Enhancer-binding Protein α Mediates Induction of Hepatic Phosphoenolpyruvate Carboxykinase by p38 Mitogen-activated Protein Kinase. *Journal of Biological Chemistry* 281: 24390-24397
- Saraswathi V, Kumar N, Gopal T, Bhatt S, Ai W, Ma C, Talmon GA, Desouza C (2020) Lauric Acid versus Palmitic Acid: Effects on Adipose Tissue Inflammation, Insulin Resistance, and Non-Alcoholic Fatty Liver Disease in Obesity. *Biology* 9: 346
- Schönfeld P, Wojtczak L (2016) Short- and medium-chain fatty acids in energy metabolism: the cellular perspective. *Journal of Lipid Research* 57: 943-954
- Soupe E, Kuypers FA (2008) Mammalian Long-Chain Acyl-CoA Synthetases. *Experimental Biology and Medicine* 233: 507-521
- Turner N, Hariharan K, TidAng J, Frangioudakis G, Beale SM, Wright LE, Zeng XY, Leslie SJ, Li J-Y, Kraegen EW, Cooney GJ, Ye J-M (2009) Enhancement of Muscle Mitochondrial Oxidative Capacity and Alterations in Insulin Action Are Lipid Species Dependent. *Diabetes* 58: 2547-2554
- Wang J, Ma J, Nie H, Zhang XJ, Zhang P, She ZG, Li H, Ji YX, Cai J (2020) Hepatic Regulator of G Protein Signaling 5 Ameliorates Nonalcoholic Fatty Liver Disease by Suppressing Transforming Growth Factor Beta-Activated Kinase 1-c-Jun-N-Terminal Kinase/p38 Signaling. *Hepatology* 73: 104-125
- Wang X, Li H, Chen Y, Fu J, Ren Y, Dong L, Tang S, Liu S, Wu M, Wang H (1992) p28GANK knockdown-derived reactive oxygen species induces apoptosis through mitochondrial dysfunction mediated by p38 in hepG2 cells. *International Journal of Oncology* 33: 743-750
- Watkins PA, Maiguel D, Jia Z, Pevsner J (2007) Evidence for 26 distinct acyl-coenzyme A synthetase genes in the human genome. *Journal of Lipid Research* 48: 2736-2750
- Woo JH, Seo HJ, Lee JY, Lee I, Jeon K, Kim B, Lee K (2023) Polypropylene nanoplastic exposure leads to lung inflammation through p38-mediated NF-kappaB pathway due to mitochondrial damage. *Part Fibre Toxicol* 20: 2

- Wu T, Liu Q, Li Y, Li H, Chen L, Yang X, Tang Q, Pu S, Kuang J, Li R, Huang Y, Zhang J, Zhang Z, Zhou J, Huang C, Zhang G, Zhao Y, Zou M, Jiang W, Mo L et al. (2021) Feeding-induced hepatokine, Manf, ameliorates diet-induced obesity by promoting adipose browning via p38 MAPK pathway. *Journal of Experimental Medicine* 218: e20201203
- Yu Y, Rajapakse AG, Montani J-P, Yang Z, Ming X-F (2014) p38 mitogen-activated protein kinase is involved in arginase-II-mediated eNOS-Uncoupling in Obesity. *Cardiovascular Diabetology* 13: 113
- Zhang X, Fan L, Wu J, Xu H, Leung WY, Fu K, Wu J, Liu K, Man K, Yang X, Han J, Ren J, Yu J (2019) Macrophage p38 α promotes nutritional steatohepatitis through M1 polarization. *Journal of Hepatology* 71: 163-174

Dear Dr Wang,

Thank you for submitting your revised manuscript (EMBOJ-2023-114416R) to The EMBO Journal. Please accept my apologies for the unusual delay in assessing your amended manuscript, which was due to protracted referee input as well as detailed discussion in the editorial team. Your amended study was sent back to the referees for their re-evaluation, and we have received comments from both, which I enclose below.

As you will see, the experts stated that the work has been substantially improved by the revisions and they are now in broadly favour of publication.

Thus, we are pleased to inform you that your manuscript has been accepted in principle for publication in The EMBO Journal.

We now need you to take care of a number of minor issues related to formatting and data presentation as detailed below, which should be addressed at re-submission.

Please contact me at any time if you have additional questions related to below points.

As you might have noted on our web page, every paper at the EMBO Journal now includes a 'Synopsis', displayed on the html and freely accessible to all readers. The synopsis includes a 'model' figure as well as 2-5 one-short-sentence bullet points that summarize the article. I would appreciate if you could provide this figure and the bullet points.

Thank you for giving us the chance to consider your manuscript for The EMBO Journal. I look forward to your final revision.

Again, please contact me at any time if you need any help or have further questions.

Kind regards,

Daniel Klimmeck

>> Please limit the keywords for your study to maximally five.

>> Merge the current 'Conflict of Interests' and 'Disclosures' sections and adjust the title to 'Disclosure and Competing Interests Statement'.

>> Author Contributions: Please remove the author contributions information from the manuscript text. Note that CRediT has replaced the traditional author contributions section as of now because it offers a systematic machine-readable author contributions format that allows for more effective research assessment. and use the free text boxes beneath each contributing author's name to add specific details on the author's contribution.

More information is available in our guide to authors.
<https://www.embopress.org/page/journal/14602075/authorguide>

>> Order manuscript: the Material and Methods part should be after Discussion

>> Dataset EV legends: EV table and EV dataset need titles added to the legends in the excel files-.

>> Funding information: please merge the information with the 'Acknowledgements' section.

>> Callouts: need to be adjusted to the correct sequence - Fig 2A,B ; Fig 3J,K ; Fig 4A and Fig 5A should be mentioned after Figure 1.

>> Data availability section: Please provide the specific URL for GSE247290 dataset in the data availability statement.

>> Indicate information regarding consent and ethics committee approval for the second patient cohort in the Material and Methods section.

>> Consider additional changes and comments from our production team as indicated below:

- Figure legends:

1. Please note that the figure legend style does not comply with the journal guidelines i.e. all the figure legends are in a run-on style.
2. Please note that a separate 'Data Information' section is required in the legends of all the figures.
3. Please note that the box plots need to be defined in terms of minima, maxima, centre, bounds of box and whiskers, and percentile in the legend of figure 1c.
4. Please note that information related to n is missing in the legend of figure 1c

Referee #1:

The authors have addressed all comments, and I am satisfied with the revised manuscript. It should be highlighted that inclusion of new data demonstrating that Acsm3 is predominantly expressed in mouse liver and downregulated in MetS mice is compelling and has strengthened the significance of the study.

Referee #2:

The Authors have carefully and thoroughly responded to all the concerns I raised in my initial review. The rationale of the paper is now much easier to follow, the experiments provide the proper background and controls, and the rewritten discussion is appropriate in length and scope. I have no further comments or concerns with this manuscript. As I mentioned in my initial Review, the findings of this paper are of potentially very high impact, and have the chance to drive a new direction in the field of research into metabolic syndrome.

The authors addressed the minor editorial issues.

Dear Dr Wang,

Thank you for submitting the revised version of your manuscript. I have now evaluated your amended manuscript and concluded that the remaining minor concerns have been sufficiently addressed.

Thus, I am pleased to inform you that your manuscript has been accepted for publication in the EMBO Journal.

Please note that it is The EMBO Journal policy for the transcript of the editorial process (containing referee reports and your response letter) to be published as an online supplement to each paper.

If you do NOT want the transparent process file published, you will need to inform the Editorial Office via email immediately. More information is available here: https://www.embopress.org/transparent-process#Review_Process

On a different note, I would like to alert you that EMBO Press offers a format for a video-synopsis of work published with us, which essentially is a short, author-generated film explaining the core findings in hand drawings, and, as we believe, can be very useful to increase visibility of the work. This has proven to offer a nice opportunity for exposure i.p. for the first author(s) of the study. Please see the following link for representative examples and their integration into the article web page:

<https://www.embopress.org/doi/full/10.15252/emj.2019103932>

If you have any questions, please do not hesitate to call or email the Editorial Office.

Kind regards,

Daniel Klimmeck

Daniel Klimmeck, PhD
Senior Editor
The EMBO Journal
EMBO
Postfach 1022-40
Meyerhofstrasse 1
D-69117 Heidelberg
contact@embojournal.org
Submit at: <http://emboj.msubmit.net>